# Inhalation of ACE2-expressing lung exosomes provides prophylactic protection against SARS-CoV-2

Zhenzhen Wang [1,2,3] ✉, Shiqi Hu [4], Kristen D. Popowski [2,3], Shuo Liu[4], Dashuai Zhu [4], Xuan Mei[2,3], Junlang Li[5], Yilan Hu [4], Phuong-Uyen C. Dinh [2,3], Xiaojie Wang[6,7] ✉ & Ke Cheng [4] ✉

Continued emergence of SARS-CoV-2 variants of concern that are capable of escaping vaccine-induced immunity highlights the urgency of developing new COVID-19 therapeutics. An essential mechanism for SARS-CoV-2 infection begins with the viral spike protein binding to the human ACE2. Consequently, inhibiting this interaction becomes a highly promising therapeutic strategy against COVID-19. Herein, we demonstrate that ACE2-expressing human lung spheroid cells (LSC)-derived exosomes (LSC-Exo) could function as a prophylactic agent to bind and neutralize SARS-CoV-2, protecting the host against SARS-CoV-2 infection. Inhalation of LSC-Exo facilitates its deposition and biodistribution throughout the whole lung in a female mouse model. We show that LSC-Exo blocks the interaction of SARS-CoV-2 with host cells in vitro and in vivo by neutralizing the virus. LSC-Exo treatment protects hamsters from SARS-CoV-2-induced disease and reduced viral loads. Furthermore, LSC-Exo intercepts the entry of multiple SARS-CoV-2 variant pseudoviruses in female mice and shows comparable or equal potency against the wild-type strain, demonstrating that LSC-Exo may act as a broad-spectrum protectant against existing and emerging virus variants.

The coronavirus disease 2019 (COVID-19) pandemic, provoked by severe acute respiratory syndrome coronavirus 2 (SARS-CoV-2), presents a global health crisis to the public and has to date resulted in over 772 million infections and more than 6.9 million deaths[1]. Successful development of vaccines that possess reported efficacy rates up to 95% has reduced the COVID-19 morbidity and mortality[2,3]. However, an increasing number of SARS-CoV-2 variants of concern (VOC) have been identified globally[4]. Spike (S) protein undergoes mutations at all times to optimize its binding mode and affinity to human angiotensin-converting enzyme II (hACE2) receptors[5]. Such mutations not only altered SARS-CoV-2 pathogenesis, virulence, and transmissibility, but importantly raised severe concerns regarding current vaccines' effectiveness against mutated viruses[6]. Some variants, including B.1.1.7 (Alpha), B.1.617.2 (Delta), and B.1.1.529 (Omicron) variants were highly resistant to BNT162b2 or mRNA-1273 vaccine-induced humoral immunity[7–9]. As such, it is becoming undeniable evident that developing innovative and cost-effective interventions is necessary to prevent infection by SARS-CoV-2 variants, ideally providing prophylaxis at

[1]School of Biomedical Sciences and Engineering, South China University of Technology, Guangzhou International Campus, Guangzhou 511442, P.R. China. [2]Department of Molecular Biomedical Sciences, North Carolina State University, Raleigh, NC 27606, USA. [3]Joint Department of Biomedical Engineering, University of North Carolina at Chapel Hill, Chapel Hill, North Carolina 27599, and North Carolina State University, Raleigh, NC 27606, USA. [4]Department of Biomedical Engineering, Columbia University, New York, New York 10032, USA. [5]Xsome Biotech Inc., Raleigh, North Carolina 27607, USA. [6]School of Pharmacy, Wenzhou Medical University, Wenzhou 325035, P.R. China. [7]Engineering Research Center of the Chinese Ministry of Education for Bioreactor and Pharmaceutical Development, Jilin Agricultural University, Changchun 130118, P.R. China. ✉e-mail: zwang@scut.edu.cn; susanwang1214@wmu.edu.cn; ke.cheng@columbia.edu

the virus entry portal and disease progression, which limits the virus-induced damage and transmission.

Given that the infectivity of SARS-CoV-2 relies on binding its S protein with the entry receptor hACE2, inhibiting this interaction is therefore a promising treatment strategy[10–12]. Potent neutralizing monoclonal antibodies (mAbs) have been developed to target S protein for harnessing endogenous host defense mechanisms against SARS-CoV-2 infection[13–15]. However, mAbs typically exhibit reduced neutralization capacity against many variants[16]. While there is substantial evidence indicating that mAbs did not induce antibody-dependent enhancement (ADE) effect in vivo, recent studies show that several antibodies, such as XG016, XG005, DH1047, DH1041, and MW05, did, indeed, induce ADE, using either pseudoviruses or authentic viruses[17]. Since SARS-CoV-2 VOC harbor mutations that could increase virus attachment to ACE2 receptor and coronavirus lineages appear to exhibit a stronger affinity for docking on ACE2 receptors, creating and employing ACE2 decoys might inhibit SARS-CoV-2 infectivity, raising the possibility of combatting any future variants. Particularly, intravenously injected recombinant hACE2 protein (rhACE2) as biological therapeutics has been developed and showed great potential to intercept the entry of virus, limit the progression of infection and reduce lung injury[18,19]. However, the rapid degradation of free rhACE2 and the notoriously low efficiency of intravascular delivery across the plasma-lung barrier would greatly hamper their therapeutic efficacy against pulmonary infections.

Recent studies hint at the potential of cellular membrane-derived nanovesicles (NVs) displaying hACE2 that compete with host cells for SARS-CoV-2 binding, protecting the host cells against SARS-CoV-2 infection[20–22]. Our laboratory has developed hACE2 NVs derived from healthy human lung spheroid cells (LSC) that could serve as decoys to neutralize SARS-CoV-2 and trigger subsequent phagocytosis by macrophages to clear the virus in a non-human primate model[23]. Additionally, engineered extracellular vesicles with enriched hACE2 expression have been demonstrated to efficiently protect mice against SARS-CoV-2 lung inflammation[24,25]. Although promising, their further clinical translations were hindered by the random orientations of hACE2 on cell membrane-derived NVs and the potential risks of gene engineering[26,27]. Interestingly, exosomes and viruses employ similar endosomal sorting pathways and mechanisms, endowing exosomes with the potency to be a new therapeutic reagent for targeting, binding, and suppressing cellular uptake of various viruses including SARS-CoV-2[28,29]. Furthermore, by sharing surface receptor proteins, microRNA, and DNA with their parental cells, lung-derived exosomes would harness superior homing-target ability towards lung over their exogenous counterparts[30,31]. Our laboratory has successfully developed LSC as a cell therapy from initial rodent studies to an ongoing phase 1 clinical trial (NCT04262167)[32]. LSC represent natural mixtures of resident lung epithelial cells consisting of both types I and II pneumocytes and mesenchymal cells. Being resident lung cells, they express ACE2 naturally, and we therefore speculate that LSC-derived exosomes (LSC-Exo) could carry the parental cell's ACE2, target lung, and confer protection against SARS-CoV-2 (Fig. 1a)

In this study, we systematically assess the efficacy of LSC-Exo for prophylactic protection against SARS-CoV-2 infection and SARS-CoV-2 B.1.617.2 (Delta) variant. Since SARS-CoV-2 infection typically begins in the nasal cavity, followed by aspiration of the viral inoculum from the oropharynx into the lower respiratory tract[33,34], inhalation delivery of LSC-Exo is performed to endow effective protective benefits on the affected sites. We provide direct evidence that LSC-Exo expresses enriched hACE2 and can cross the air-blood-barrier to reach and accumulate in trachea, bronchioles, and deep lung parenchyma after nebulization. Importantly, LSC-Exo significantly prevents SARS-CoV-2 infection in Syrian hamsters, a model of severe COVID-19 disease[35], by a drastically reduced viral load, diminished lung inflammation, and dampened viral pneumonia. More importantly, we demonstrate that LSC-Exo preserves the neutralizing capacity against D614G and B.1.617.2 (Delta) pseudoviruses. These natural LSC-Exo shows great potential for blocking the entry of SARS-CoV-2 variants into host cells and may serve as a daily prophylaxis reagent to offer the necessary protection against the infection by SARS-CoV-2 variants.

## Results

### Characterization of LSC-Exo

ACE2 levels in LSC were analyzed by immunofluorescent imaging (Fig. 1b) and immunoblotting (Figs. 1c, S1), in which HEK293T cells (HEK) with low ACE2 expression were used as a negative control. These results demonstrated that LSC showed 6-fold higher ACE2 expression levels than HEK (Fig. 1c). Confocal imaging identified that those ACE2 receptors were present on the membrane of two subpopulations of LSC, surfactant-associated protein c-positive (SFTPC+)-Type II pneumocytes and aquaporin 5-positive (AQP5+) Type I pneumocytes (Fig. S2), which consistent with previous studies[32]. Both LSC-derived and HEK-derived extracellular vesicles were collected and purified. In our previous works, LSC-derived extracellular vesicles was termed as LSC-exosome (LSC-Exo). Note that the term 'small extracellular vesicles' is considered more accurate than exosomes for characterizing the purified LSC-derived extracellular vesicles according to the MISEV guidelines[36]. We found that LSC-Exo and HEK-Exo exhibited a similar morphology and size as measured by transmission electron microscopy (TEM, Fig. 1d) and nanoparticle tracking analysis (NTA, Fig. 1e). In addition, direct stochastic optical reconstruction microscopy (dSTORM) imaging suggested that distinct CD9, CD63 and CD81 surface biomarkers were detected on single LSC-Exo and HEK-Exo. Western blot further demonstrated that the expression of cytosolic marker TSG101 on both LSC-Exo and HEK-Exo compared to the negative biomarker of prohibitin (Fig. S3). Mass spectrometry analysis was performed to examine LSC-Exo's proteome. Venn diagram and correlation scatterplots analysis revealed that LSC-Exo and HEK-Exo shared 2146 proteins (Fig. S4a, b). We identified the expression of CD9, CD63, CD81, TSG101, Alix, and VSP36 biomarkers in both LSC-Exo and HEK-Exo (Fig. S4c). Gene Ontology (GO) function and Kyoto encyclopedia of genes and genomes (KEGG) analysis indicated LSC-Exo was mainly involved in the extracellular matrix organization, response to growth factor and response to wounding through cytokine-cytokine receptor interaction, TGF-β and NOD-like receptor signaling pathway etc. (Fig. S4d, e). We further sought to evaluate the level of ACE2 on LSC-Exo and HEK-Exo by flow cytometry and immunoblotting assay. We found that LSC-Exo exhibited significantly higher ACE2 expression than HEK-Exo (Fig. 1f), in line with the results from their parent cells. Immunoblotting assays further validated that LSC-Exo, but not HEK-Exo, expressed enriched hACE2 (Fig. 1g).

### Biodistribution of LSC-Exo in mice after nebulization

We set out to study the biodistribution and retention of LSC-Exo in rodent lungs after nebulization and compared it with the gold-standard delivery vesicle liposome. To visualize the LSC-Exo in vivo, red fluorescent proteins (RFP) were loaded into LSC-Exo (RFP-Exo), HEK-Exo (RFP-HEK) and commercially available liposomes (RFP-Lipo) by electroporation, respectively. After inhalation of LSC-Exo, healthy mice were sacrificed 2-or 4- or 24-hours later (Fig. 2a). Ex vivo imaging (Fig. 2b) and analysis (Fig. 2c) of the murine lung exhibited the greatest RFP integrated density in mice which were sacrificed after 2 h. We found that significantly more LSC-Exo was retained and distributed throughout the whole lung than liposomes over time, despite the similar distribution of both at 2 h post-inhalation (Fig. 2d). These results were consistent with our previous report[37]. Notably, significantly more LSC-Exo reached the trachea than liposomes (Fig. 2e). Both bronchioles and parenchyma began to show LSC-Exo signals 2 h after inhalation, indicating that LSC-Exo crossed the air-blood-barrier to reach and accumulate the lung of mice. Significantly, fewer liposomes were observed

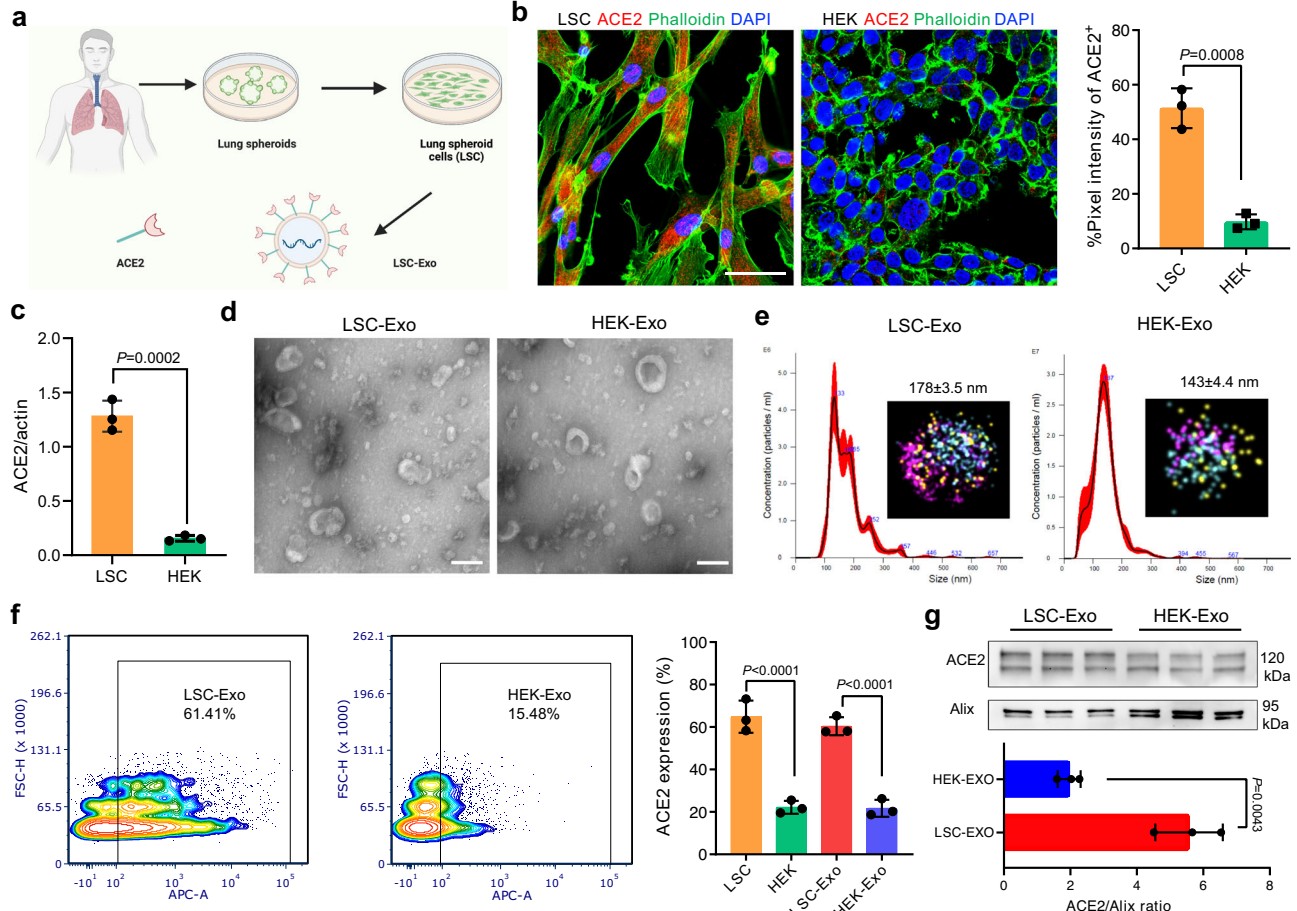

**Fig. 1 | Characterization of LSC-Exo and surface expression of ACE2 on LSC-Exo. a** Extraction scheme of LSC and LSC-Exo from healthy donors, created with Biorender.com. **b** Immunofluorescence staining and quantification analysis of ACE2 on LSC and HEK. Scale bar: 50 μm. *n* = 3. **c** Western blot quantification of ACE2 expression in LSC and HEK, which derived from the same experiments and processed in parallel. *n* = 3. **d** Representative TEM images of LSC-Exo and HEK-Exo from 3 independent experiments. Scale bar: 100 μm. **e** Measurements of size distribution of LSC-Exo and HEK-Exo via nanoparticle tracking analysis. Inset: 3-colar dSTORM image of CD63-Alexa Fluor®-488, PE-CD9, APC-CD81 of LSC-Exo or HEK-Exo. **f** Quantification of ACE2 expression on LSC-Exo and HEK-Exo by flow cytometry. *n* = 3. Gating strategy was shown in Fig. S20a. **g** Western blot and quantification analysis of ACE2 levels on LSC-Exo and HEK-Exo. This quantification analysis derived from the same experiments. *n* = 3. Data are mean ± s.d. A two-tailed, unpaired Student's *t*-test was performed for statistical analysis. Source data are provided as a Source Data file.

to reach the deep bronchioles and parenchyma than LSC-Exo, indicative of rapid degradation and clearance of liposomes in mice. In addition, we found that HEK-Exo signals exhibited a distribution similar to that of LSC-Exo at both 2 h and 4 h post-inhalation (Fig. 2b–d). However, after 24 h of inhalation, fewer HEK-Exo signals were detected in trachea, bronchioles and parenchyma of lungs compared to LSC-Exo (Fig. 2e), indicating that LSC-Exo exhibited a longer retention time. These datasets suggested that inhalation is an effective route for pulmonary delivery of LSC-Exo to murine.

We further studied whether LSC-Exo could be delivered into other organs of mice via inhalation administration. In addition to lung, we found that both HEK-Exo and liposome signals began to appear in liver, spleen, and kidneys after 4 h inhalation (Fig. S5a, b). Comparatively, LSC-Exo signals began to appear in major organs at 24 h post-inhalation, which could be attributed to the homologous targeting effects of LSC-Exo to lungs. Confocal images of 24 h post-inhalation exhibited the obvious LSC-Exo signals in the heart, liver, spleen and kidneys (Fig. S5c).

## LSC-Exo neutralizes SARS-CoV-2 pseudovirus in vitro and in vivo

To evaluate the neutralizing activity of LSC-Exo against original SARS-CoV-2 WA1 pseudovirus, we sought to study the binding

affinity of LSC-Exo with receptor binding domain (RBD) of S protein by biolayer interferometry (BLI) analysis. At equivalent concentrations, LSC-Exo was found to bind to RBD protein similarly to free rhACE2, and more strongly than HEK-Exo did (Fig. 3a). ELISA-based blocking assay further confirmed that LSC-Exo inhibited the specific binding of RBD with rhACE2 in a dose-dependent manner, unlike HEK-Exo (Fig. 3b), indicating that LSC-Exo has a stronger binding ability to RBD. To analyze the efficacy of LSC-Exo on viral attachment and infection, we implemented a SARS-CoV-2 pseudovirus-based assay assessing the protective activity of LSC-Exo to A549 cells expressing ACE2 receptor (Fig. 3c). In a dose-dependent manner, LSC-Exo efficiently intercepted the entry of SARS-CoV-2 pseudovirus with an GFP reporter into ACE2-expressing A549 cells (Fig. 3d, e). In contrast, an equal amount of HEK-Exo had negligible inhibition effects, whereas the positive control, rhACE2, efficiently blocked the infection of SARS-CoV-2 pseudovirus in A549 cells. Flow cytometry (Fig. 3f) and confocal imaging (Fig. S6) further validated that LSC-Exo and rhACE2 efficiently neutralized SARS-CoV-2 pseudoviruses and prevented them entry into host cells, while HEK-Exo failed to inhibit this entry.

Having demonstrated that LSC-Exo was able to neutralize SARS-CoV-2 at the cellular level, we further evaluated its neutralization

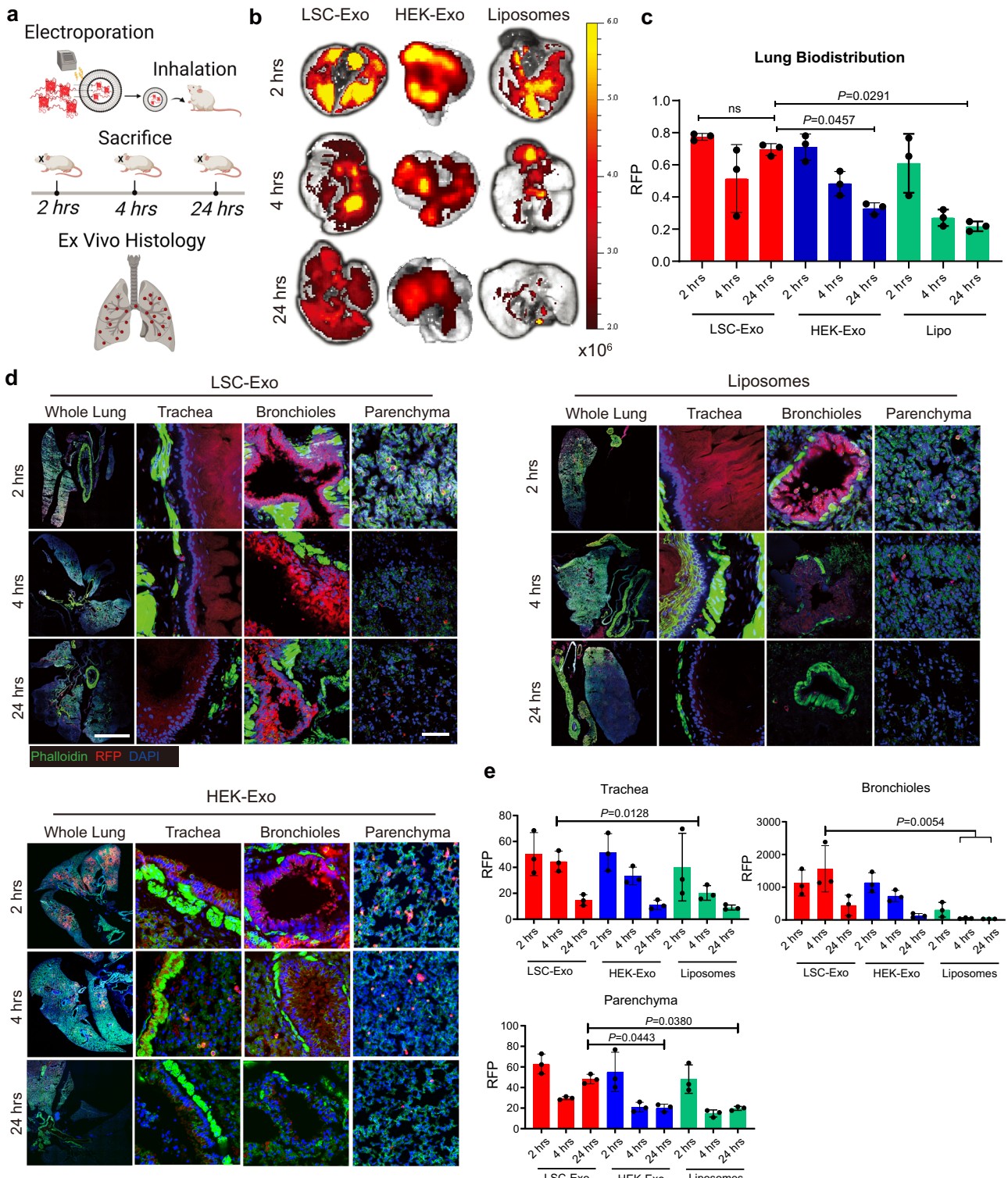

**Fig. 2 | RFP-loaded LSC-Exo has superior distribution in mouse lung. a** Study scheme of RFP-Exo's distribution in mice, created with Biorender.com. **b** 2, 4 and 24 h ex-vivo imaging of mouse lungs after RFP-Exo, RFP-HEK or RFP-Lipo neb-ulization. **c** Quantitative results of RFP intensity in ex-vivo mouse lungs; $n = 3$. **d** Representative immunostaining images for whole lung, trachea, bronchioles, and parenchyma. Exosomes (red) or liposomes (red), phalloidin (green) and DAPI (blue). Scale bar, 1000 µm for whole lung, 100 µm for parenchyma. **e** Quantification results of RFP fluorescence density in trachea, bronchioles, and parenchyma; $n = 3$ per group. Data are mean ± s.d. The one-way ANOVA with Bonferroni correction was performed for statistical analysis. Source data are provided as a Source Data file.

ability in vivo. On the basis of the results of LSC-Exo's biodistribution in vivo, mice were nebulized with LSC-Exo 2 h before SARS-CoV-2 pseudovirus challenge (Fig. 3g). Ex-vivo fluorescence imaging showed that substantial pseudovirus signals were detected in the mice treated

with HEK-Exo (Fig. 3h). Conversely, dim pseudovirus signals were observed in the mice treated with LSC-Exo, indicative of successful inhibition of virus entry. Intriguingly, rhACE2 failed to block SARS-CoV-2 pseudovirus entry into mice (Fig. 3h), which might be attributed to

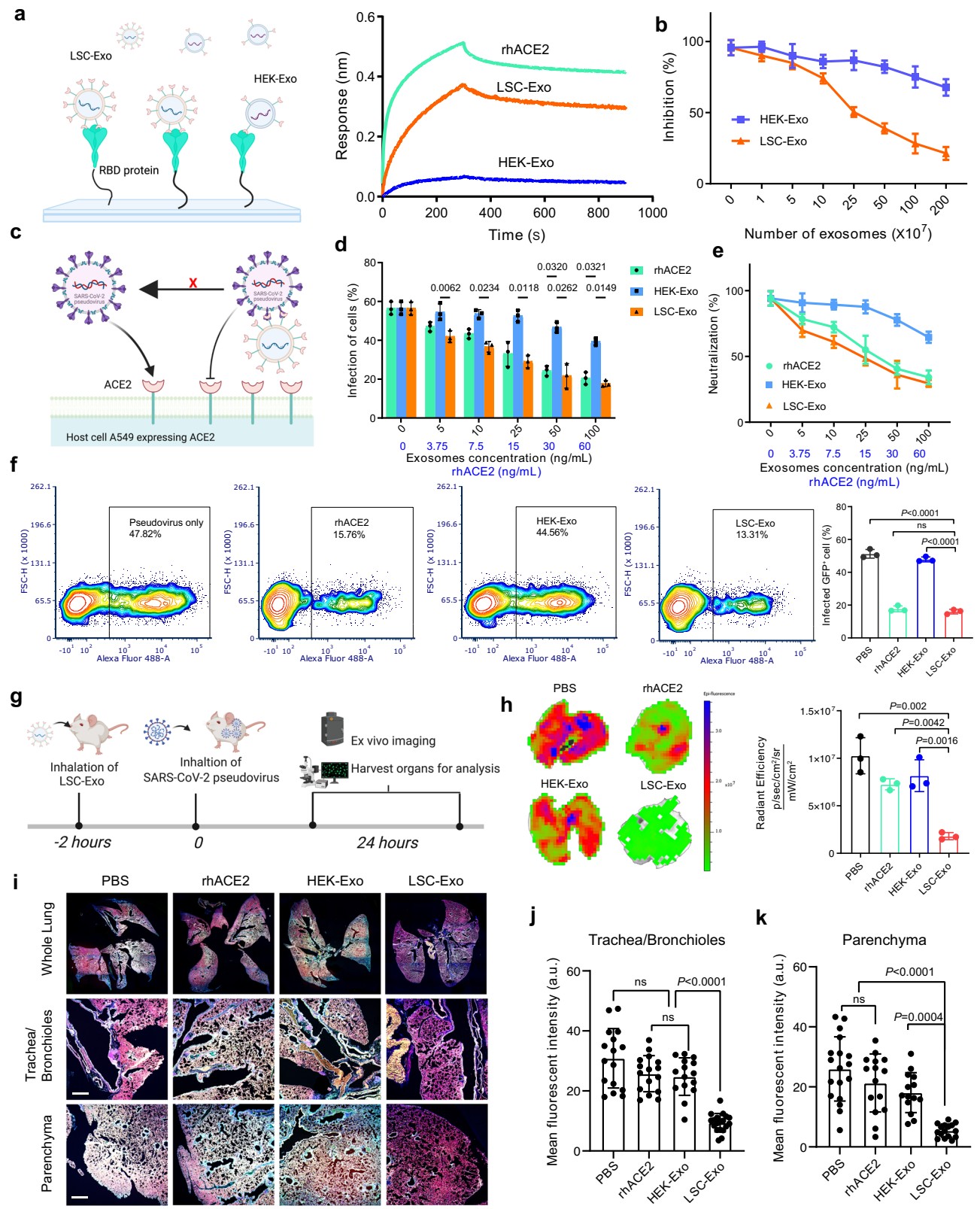

the rapid clearance of free rhACE2 in vivo. Whole lung imaging further confirmed that fewer SARS-CoV-2 pseudoviruses were distributed in both the trachea/bronchioles and parenchyma in the mice with LSC-Exo treatment, rather than HEK-Exo or rhACE2 treatment (Fig. 3i–k). Collectively, those compound datasets suggested that LSC-Exo is capable of neutralizing the SARS-CoV-2 pseudovirus and preventing their infection to the host cells.

## LSC-Exo protects Syrian hamsters from SARS-CoV-2 infection

Syrian golden hamsters, exhibiting diverse pathologies characteristic of SARS-CoV-2 infection[38], were utilized to evaluate the prophylactic and therapeutic capacity of LSC-Exo against original SARS-CoV-2 WA1 infection. Prior to that, we evaluated the delivery of LSC-Exo into hamsters following inhalation administration. We observed that LSC-Exo was predominantly accumulated in the lungs of hamsters 2 h after

**Fig. 3 | LSC-Exo prevents the entry of SARS-CoV-2 pseudovirus. a** Biolayer interferometry assay of the binding LSC-Exo or HEK-Exo or rhACE2 to RBD. The left panel was created with Biorender.com. **b** ELISA analysis of the binding affinity between rhACE2 with RBD in the presence of LSC-Exo or HEK-Exo. $n = 3$. **c** Schematic depiction of cell-based neutralization assay, created with Biorender.com. **d** SARS-CoV-2 WA1 pseudovirus neutralization analysis of LSC-Exo, HEK-Exo, or rhACE2 in A549 cells expressing ACE2, determined by GFP fluorescence intensity. $n = 3$. **e** The neutralization potency of LSC-Exo determined by SARS-CoV-2 WA1 pseudovirus neutralization analysis. $n = 3$. **f** Flow plots of SARS-CoV-2 WA1 pseudovirus-infected A549 cells that inhibited by LSC-Exo, HEK-Exo, or rhACE2 and its corresponding quantification analysis. $n = 3$. Gating strategy was shown in Fig. S20b. **g** Animal study design of the protection of LSC-Exo against SARS-CoV-2 WA1 pseudovirus with a GFP reporter, created with Biorender.com. **h** Ex-vivo imaging and quantification analysis of lung from mice inoculated with SARS-CoV-2 WA1 pseudovirus. $n = 3$. **i** Immunostaining images of whole lung of mice for DAPI (blue), phalloidin (red), and SARS-CoV-2 WA1 pseudovirus (green). Scale bar: 50 μm. Quantification analysis of GFP reporter signals in trachea/bronchioles (**j**) and parenchyma (**k**). $n = 10$ images from 5 hamsters. Data are mean ± s.d. Statistical analysis was performed by two-way ANOVA with Tukey's multiple comparisons (**d**) or one-way ANOVA with Bonferroni correction (**f, h, j, k**). Source data are provided as a Source Data file.

inhalation (Fig S7). In contrast, after 24 h of inhalation, LSC-Exo exhibited substantial distribution throughout the major organs of the hamsters. These results were consistent with the biodistribution results of LSC-Exo in mice.

We next demonstrated that inhalation of LSC-Exo at 2 h before challenging with authentic SARS-CoV-2 WA1, significantly prevented SARS-CoV-2-induced weight loss as compared to both HEK-Exo and PBS treatment (Fig. 4a, b). Moreover, this protection was associated with decreased viral load in both oral swabs (OS) and bronchoalveolar lavage (BAL) of hamsters (Fig. 4c, d). In situ RNA hybridization analysis (RNAscope) further revealed that LSC-Exo prophylaxis resulted in less viral RNA presented in the lung tissues of hamsters compared with HEK-Exo treatment (Fig. 4e). Immunohistochemistry (IHC) staining for nucleocapsid (N) protein of the SARS-CoV-2 (SARS-N) indicated that viral protein in the lung tissues was reduced by LSC-Exo treatment relative to HEK-Exo control (Fig. 4e, f). Examination of lung tissues from infected hamsters with PBS or HEK-Exo treatment revealed swollen alveolar lining cells, remarkable inflammatory infiltrates filled with large numbers of neutrophils, macrophages, and lymphocytes in the alveolar walls and air spaces (Fig. 4g, j). Conversely, LSC-Exo treatment greatly reduced the severity and incidence of alveolar infiltration and interstitial pneumonia in hamsters. Masson's trichrome staining and Ashcroft score analysis exhibited that LSC-Exo significantly dampened lung fibrosis with the preservation of alveolar epithelial structures as compared to HEK-Exo or PBS treatment (Fig. 4h, i). Finally, we observed that both viral genomic RNA levels (Fig. 4k) and subgenomic RNA (sgRNA) levels (Fig. 4l) in the heart, liver, spleen, kidneys, and lymph nodes tissues were greatly decreased in hamsters that received LSC-Exo treatment, not HEK-Exo, indicating that LSC-Exo could protect the distant tissues of hamsters against SARS-CoV-2 infection, including but not limited to protecting the lung.

## LSC-Exo alleviates the immune response to SARS-CoV-2 infection in hamsters

An exaggerated pro-inflammatory host response to SARS-CoV-2 infection contributes to pulmonary pathology and the development of respiratory distress in a percentage of COVID-19 patients[39]. We observed that SARS-N positive cells were frequently accompanied by massive inflammatory infiltrates of activated ionized calcium binding adaptor (Iba-1⁺) and generally co-localized with pan-cytokeratin (pan-CK) cell marker (Fig. 5a), indicating that SARS-CoV-2 infected the alveolar epithelial cells. Compared to the PBS group, the dense inflammatory infiltrates were decreased by LSC-Exo treatment as validated by the lower expression of endogenous myeloperoxidase staining (MPO) in lung tissues (Fig. 5b). Furthermore, the down-regulation of an interferon-induced GTP-binding protein (MX1) with antiviral activity against a wide variety of RNA viruses was observed in hamsters which inhaled LSC-Exo, rather than PBS group (Fig. 5c), indicating that the reduced virus replication due to the protective activity of LSC-Exo. Furthermore, we demonstrated that clinical chemistry parameters of hamsters treated with LSC-Exo remained within normal ranges (Fig. S8).

To elucidate the underlying protection mechanisms, the bulk RNA sequencing (RNA-seq) analysis on lung tissues of infected hamsters that received PBS or LSC-Exo treatment was performed. In comparison with healthy sham hamster, PBS treatment to infected hamsters resulted in 3305 up-regulated genes and 3764 down-regulated genes, respectively (Fig. S9a). In stark contrast, only a handful of differentially expressed genes were detected in infected hamsters who were inhaled with LSC-Exo, where 745 and 545 genes were upregulated and downregulated, respectively (Fig. S9b). Principal component analysis (PCA) and Pearson correlation analysis results exhibited a high degree of similarity in the transcriptomic profiles between the LSC-Exo group and sham hamster, rather than PBS group with sham hamster (Fig. 5d, e). The Venn diagram in Fig. 5f revealed that 2016 unique genes were exclusively expressed in PBS group as compared to sham hamster, whereas only unique 415 differentially expressed genes were found between the LSC-Exo and sham groups. Volcano plots (Fig. 5g) showed that hundreds of upregulated genes caused by SARS-CoV-2 were downregulated by LSC-Exo treatment, many of which are associated with oxidative-reduction processes and cytokine-mediated signaling pathways. Of special note, we found that LSC-Exo was capable of upregulating the expression of *Fth1*, *Anxa5*, *Spns2*, *Emp2* and *Calm1* markers, which are beneficial for alleviating SARS-CoV-2 pathogenesis and improving pulmonary function in hamsters.

GO analysis revealed that SARS-CoV-2 infection significantly deteriorated many key biological processes of hamsters (Fig. S10). Comparatively, LSC-Exo treatment significantly restored a network of genes that center on cytokine signaling in the immune system, cytokine-cytokine receptor interaction, and regulation of cellular response to stress (Fig. 5h). In addition, GO enrichment analysis further revealed that a number of gene clusters in terms of regulation of cell adhesion, positive regulation of cell migration, and neutrophil degranulation were significantly upregulated by LSC-Exo treatment (Fig. 5h). These results indicated that LSC-Exo has the potential to dampen the systemic damage caused by SARS-CoV-2. To gain insight into the response of LSC-Exo to SARS-CoV-2 infection at the gene level, we further evaluated a panel of genes involved in inflammatory responses in greater detail (Fig. 5i–l). We found that SARS-CoV-2 disrupted the cellular redox balance of hamsters through oxidative phosphorylation pathway, in which the *Ndufb1*, *Ndufab1*, *Atp5e*, *Atp12a*, *Atp5d* genes were upregulated and *Atp6v0a1*, *Ndufs1*, and *Sdhb* genes were downregulated (Fig. 5i). These signature genes indicated the imbalance of mitochondrial electron transport chain and the dysregulation of mitochondria function[40,41]. In comparison, LSC-Exo inhalation efficiently abolished abnormal oxidative phosphorylation, of which many genes were at levels equivalent to sham hamsters (Fig. 5i). Literature has reported that mitochondria dysfunction stimulates the generation of reactive oxygen species (ROS) and consequently triggers an aberrant cytokine storm[42]. Compared with PBS treatment, LSC-Exo was able to maintain lung tissue's normal ROS metabolic process, as well as regulate both response and cellular response to ROS (Figs. S11–14). Furthermore, the TGF-β

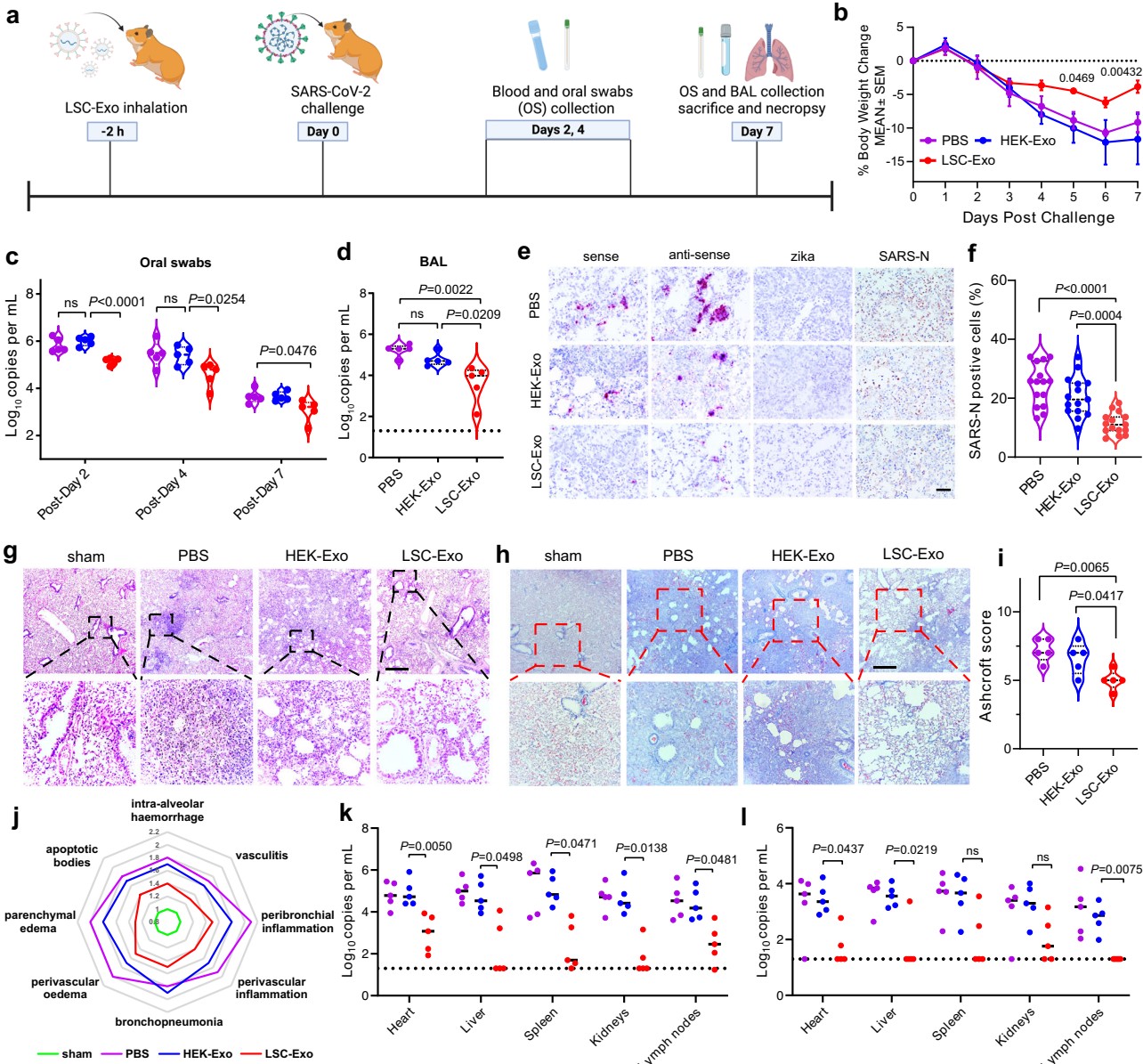

**Fig. 4 | Protective effect of LSC-Exo against authentic SARS-CoV-2 infection in Syrian hamsters. a** Time courses of LSC-Exo inhalation, viral challenge, and measurements, created with Biorender.com. **b** Changes in body weight of hamsters over 1-week post-challenge. $n = 4$. **c** Viral RNA in oral swabs (OS) from hamsters treated with LSC-Exo, HEK-Exo or PBS. $n = 5$. **d** Viral RNA in bronchoalveolar lavage (BAL) fluid from hamsters treated with LSC-Exo, HEK-Exo or PBS at 7 days post-challenge. $n = 5$. **e** RNAscope images revealing regional distribution and viral RNA levels in hamster lungs. Immunohistochemistry analysis of SARS-N protein in lung tissues of hamsters. Scale bar, 50 μm. **f** Quantification analysis of positive SARS-N cell percentages in lungs of hamster. $n = 15$. **g** H&E images of representative lung sections of hamsters. $n = 5$ animals per group. Three images were taken for each animal. Scale bar, 500 μm. **h** Masson's trichrome staining of lung sections of hamsters. $n = 5$ animals per group. Three images were taken for each animal. Scale bar, 500 μm. **i** Ashcroft scoring analysis of lung fibrosis from challenged hamsters that performed blindly. $n = 5$. **j** Spider web plot displaying histopathological scoring of lung damage, normalized to sham control (green). Viral genomic RNA levels (**k**) and sgRNA levels (**l**) in tissues of hamsters with PBS, HEK-Exo or LSC-Exo treatment (purple, PBS; blue, HEK-Exo; and red, LSC-Exo). $n = 5$. Data are mean ± s.d. Statistical analysis was performed by two-way ANOVA with Tukey's multiple comparisons (**b, c, k, l**) or one-way ANOVA with Bonferroni correction (**d, f, i**). Source data are provided as a Source Data file.

signaling, cytokine mediated signaling, NK differentiation and MAPK signaling pathway were highly associated with the therapeutic mechanisms of LSC-Exo (Fig. 5j–l). Network analyses (Fig. 5m) identified the eight modules with respect to ROS homeostasis and proved 5 key hub genes (*Cat, Foxo3, Ogt, Ncf1* and *Prcp*) that could modulate the antioxidant defense system of lung tissues against excessive oxidative stress caused by SARS-CoV-2[43–45]. According to the differential expression pattern of whole-genome analysis, LSC-Exo treatment significantly protected the hamsters against SARS-CoV-2 infection as compared to PBS treatment (Fig. 5n).

## LSC-Exo broadly neutralizes SARS-CoV-2 variants of concern pseudoviruses

SARS-CoV-2 VOC is rapidly rising in frequency. Some of these mutations confer escape from prior immunity and existing therapeutic monoclonal antibodies[46]. We tested whether LSC-Exo's prophylactic capacity was recalcitrant to mutational escape, as predicted. In A549 cells expressing ACE2 receptor, LSC-Exo was demonstrated to have a broad and strong ability in protection against both SARS-CoV-2 D614G mutation pseudovirus and B.1.617.2 (Delta) variant pseudovirus infection, whereas HEK-Exo hardly showed effect against SARS-CoV-2

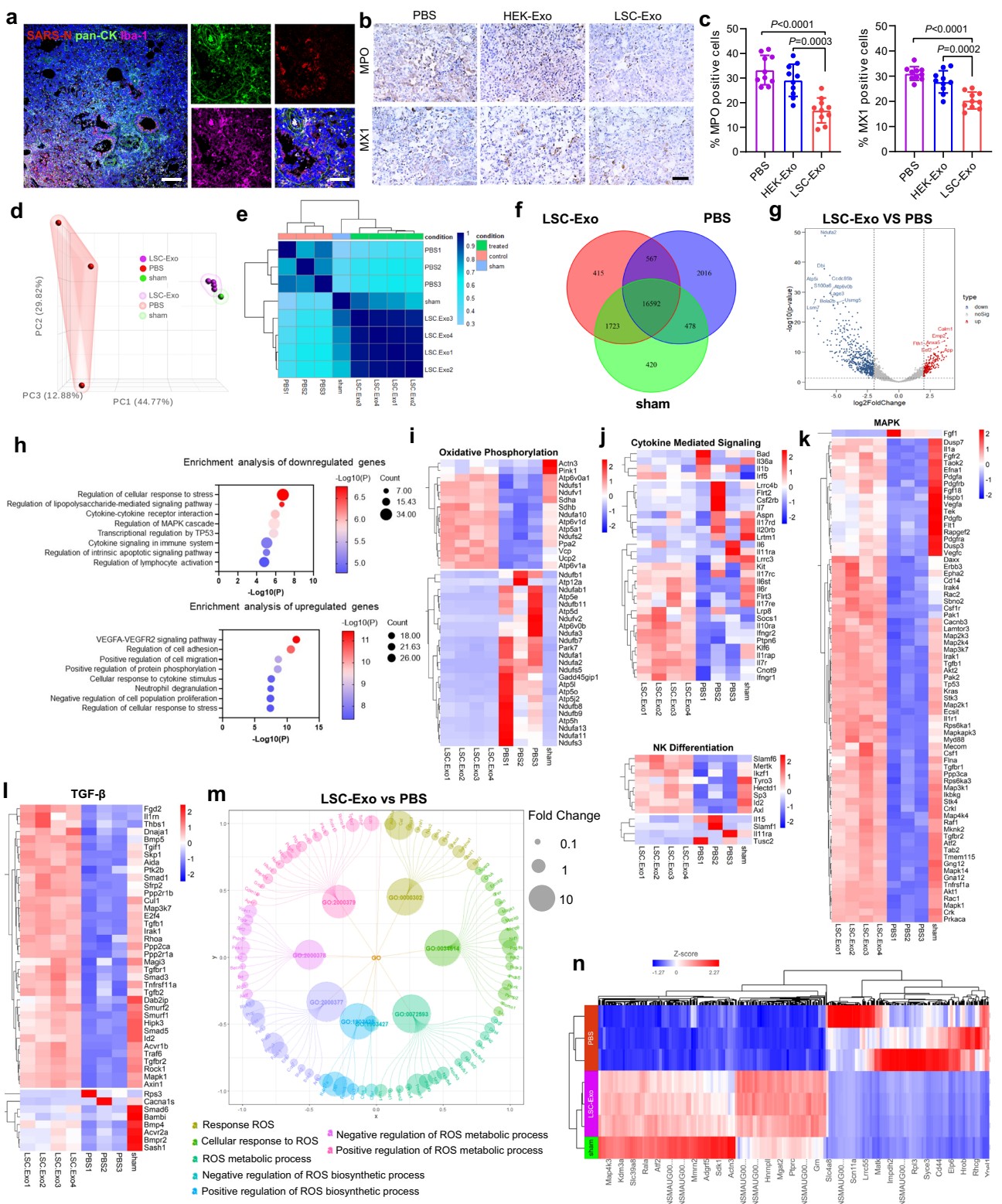

D614G and B.1.617.2 (Delta) pseudoviruses (Fig. 6a–d). The neutralization abilities of LSC-Exo was reach up to 75% for both D614G pseudovirus and B.1.617.2 (Delta) pseudovirus. Similar to the neutralization efficacy of LSC-Exo observed, free rhACE2 was capable of preventing A549 cells from SARS-CoV-2 D614G and B.1.617.2 (Delta) pseudoviruses infection at the cellular level; however, it failed to impede the entry of SARS-CoV-2 D614G and B.1.617.2 (Delta) pseudoviruses into mice. In comparison, LSC-Exo exhibited an efficient

neutralization activity in mice against both SARS-CoV-2 D614G and B.1.617.2 (Delta) pseudovirus infection, as evidenced by Ex vivo IVIS fluorescent imaging and the corresponding quantitative analysis (Fig. 6e, f). Consistently, whole lung imaging results showed that less SARS-CoV-2 D614G and B.1.617.2 (Delta) pseudoviruses were detected in the mice treated with LSC-Exo compared to those treated with HEK-Exo or rhACE2 treatment (Fig. 6g, h). Quantitative results of SARS-CoV-2 D614G and B.1.617.2 (Delta) pseudovirus signals in the whole lung

**Fig. 5 | Protective mechanisms of LSC-Exo against SARS-CoV-2 infection.**
**a** Representative SARS-N (red), pan-CK (green), Iba-1 (purple) and DAPI (blue) staining for lung tissues of hamsters. Scale bar, 50 μm. **b** Representative MPO and MX1 immunohistochemistry images from the lung sections of hamsters. Scale bar, 50 μm. **c** Quantification analysis of MPO and MX1 positive cells in hamster lungs. $n = 10$. Data are mean ± s.d. **d** Principal component analysis (PCA) comparing the transcriptome of sham hamster and infected hamsters treated with PBS or LSC-Exo. **e** Sample clustering based on Pearson's correlation of transcriptomes in lung tissues from sham, PBS and LSC-Exo group. **f** Venn diagram of the gene profiles between Sham, PBS and LSC-Exo groups. **g** Volcano plots displaying of differential gene expression from LSC-Exo versus PBS group with $P_{adj} < 0.05$, and an absolute value of $log_2$ fold change (FC) > 1 (red, upregulated genes; blue, downregulated genes). $n = 3$ for PBS group and $n = 4$ for LSC-Exo group. **h** Gene Ontology (GO) enrichment analysis of downregulated and upregulated genes from comparisons of infected hamsters treated with LSC-Exo versus PBS. Heatmaps of expression levels of candidate genes in oxidative phosphorylation (**i**), cytokine mediated signaling and NK differentiation (**j**), MAPK pathway (**k**) and TGF-β pathway (**l**) from the LSC-Exo, PBS and sham groups. **m** Functionally grouped network of enriched ROS-related categories. Each cluster is represented by a different color. **n** Heat map showing the differential gene expression of LSC-Exo vs PBS vs Sham. Statistical analysis was measured by one-way ANOVA with Bonferroni correction (**c**) or two-tailed Wald test with Benjamini–Hochberg correction for multiple comparisons (**g**, **h**). Source data are provided as a Source Data file.

imaging indicated that LSC-Exo exhibited the highest potency in neutralizing SARS-CoV-2 D614G and B.1.617.2 (Delta) pseudoviruses when compared to HEK-Exo and rhACE2, as verified by faint SARS-CoV-2 D614G and B.1.617.2 (Delta) signals observed across the trachea, bronchioles and parenchyma of the whole lung (Fig. 6i, j). Furthermore, the long-term safety of LSC-Exo was evaluated by cytokine array analysis. As illustrated in Fig. 6k and Fig. S15, compared to sham mice, no significant difference in proinflammatory cytokine expression was found in the mice treated with LSC-Exo, indicating its good safety in vivo.

### LSC-Exo treatment inhibits SARS-CoV-2 infection in hamsters
To evaluate the therapeutic capability of LSC-Exo against SARS-CoV-2 infection in hamsters, the hamsters were challenged with authentic SARS-CoV-2 WA1 firstly and then inhaled with three doses of LSC-Exo on days 1, 2, and 3 post-challenge (Fig. S16a). High levels of SARS-CoV-2 WA1 viral particles were observed in the OS for both PBS and LSC-Exo group at on day 2 post-challenge, whereas a significant decrease was observed in the LSC-Exo group at 4, 7 days post-challenge compared to PBS group (Fig. S16b). Consistent with OS results, BAL viral load was approximately 4.136 $log_{10}$ RNA copies per ml in the LSC-Exo group, which was lower than that PBS (5.23) group, indicating that LSC-Exo was capable of neutralizing SARS-CoV-2 WA1 (Fig. S16c). RNAscope analysis further demonstrated the levels of viral RNA were decreased by LSC-Exo treatment compared to PBS group (Fig. S16d). IHC analysis revealed that obvious inhibition of SARS-N expression with LSC-Exo treatment (Fig. S16e). Histological analysis revealed that SARS-CoV-2-induced pulmonary hemorrhage and edema, as well as the significant infiltration of immune cells, were effectively mitigated by LSC-Exo treatment (Fig. S16f). Furthermore, the levels of viral genomic RNA (Fig. S16g) and sgRNA (Fig. S16h) in tissues such as the heart, liver, spleen, kidneys, and lymph nodes exhibited a significant reduction in hamsters treated with LSC-Exo. The data from the PBS group presented in Fig. S16 panels (b-h) are recreated from the PBS group in Fig. 4. Moreover, clinical chemistry parameters of hamsters treated with LSC-Exo remained within the normal ranges (Fig. S17).

Clinical chemistry and complete blood count (CBC) analyses were performed to assess whether human LSC-Exo could affect the immune response of Syrian hamster when SARS-CoV-2 infection was absent. Hamsters were inhaled with LSC-Exo and sacrificed 7 days after inhalation. The clinical chemistry results indicated there was no significant difference between PBS group and LSC-Exo group (Fig. S18). CBC parameters of hamsters inhaled with LSC-Exo remained within the normal ranges, but certain CBC parameters, such as white blood cell count, neutrophil and lymphocyte count, were significantly decreased in the LSC-Exo group compared to the PBS group (Fig. S19). These results suggest that LSC-Exo might possess anti-inflammatory properties capable of diminishing the immune responses in hamsters.

### Discussion
As the critical receptor for SARS-CoV-2 entry into host cells, ACE2 took center stage in the COVID-19 outbreak, which has been demonstrated by numerous structural and biochemical interaction studies[17,47]. Accordingly, multiple drug discovery programs, including vaccine development, are focusing on the interaction of ACE2 with SARS-CoV-2 spike glycoprotein[48]. Given those characteristics, interruption of their interaction might be an promising strategy for the development of SARS-CoV-2 prophylaxis and therapeutics[49,50]. A number of neutralizing mAbs candidates or soluble rhACE2 have been developed and are currently under evaluation against SARS-CoV-2 infection in the clinic[15,51]. Furthermore, utilizing a nanodecoy with enriched ACE2 to neutralize SARS-CoV-2 and trigger the phagocytic clearance of the virus is another effective strategy for the prophylaxis and therapeutic use of COVID-19[52,53]. Although promising, the rapid degradation of free rhACE2 and the continued emergence of SARS-CoV-2 VOC greatly compromise their therapeutic efficacy. In contrast, studies have indicated that ACE2-containing defensosomes in bronchoalveolar lavage fluid from critically ill COVID-19 patients were associated with reduced intensive care unit and hospitalization times[53]. We envision that exosomes from healthy human lung or lung secretions with the inherent expression of ACE2 may provide a specialized therapeutic strategy that harbors target-homing effects and native antiviral properties, which could alleviate host inflammation and reduce viral replication in SARS-CoV-2 infection, promising for emerging SARS-CoV-2 variants.

In this study, we initially identified the expression of ACE2 receptor on healthy human lung spheroid cells (LSC)-derived exosomes (LSC-Exo) and compared them with HEK-Exo which have low ACE2 levels. We found that LSC-Exo carried much more hACE2 receptors than HEK-Exo, as verified by immunoblot analysis and flow cytometry measurements. In vitro analysis showed that LSC-Exo was able to prevent the entry of SARS-CoV-2 pseudovirus into the ACE2 receptor-expressing A549 cells, similar to free rhACE2, whereas HEK-Exo showed little neutralization capacity against SARS-CoV-2 pseudovirus. Significantly, we found that LSC-Exo exhibited a higher inhibitory activity against SARS-CoV-2 pseudovirus than rhACE2 in vivo, which could be attributed to the rapid degradation and clearance of free rhACE2 in physiological environment. Furthermore, LSC-Exo was demonstrated to be more effective in evading mucoadhesion and in directly delivering to the respiratory system over the Lipo counterpart through nebulization, suggesting that LSC-Exo has enhanced cellular targeting within the lung due to exosome phenotypes that are native to the lung microenvironment.

We systematically evaluated the prophylactic protection of LSC-Exo in SARS-CoV-2-infected Syrian hamsters, who can recapitulate serious COVID-19 disease. We found that inhalation of LSC-Exo significantly interrupted the interaction between the S protein and the entry receptor ACE2, efficiently protecting the hamsters against SARS-CoV-2 infection. In contrast to HEK-Exo group, LSC-Exo significantly decreased the viral replication as demonstrated by reduced viral load in major organ tissues of hamsters, including heart, liver, spleen, lung, kidneys, and lymph nodes. Lung examinations revealed that hamsters who were inhaled LSC-Exo did not exhibit fulminant pulmonary disease as observed in hamsters treated with HEK-Exo. RNA-Seq analysis provided direct evidence that LSC-Exo not only were able to efficiently

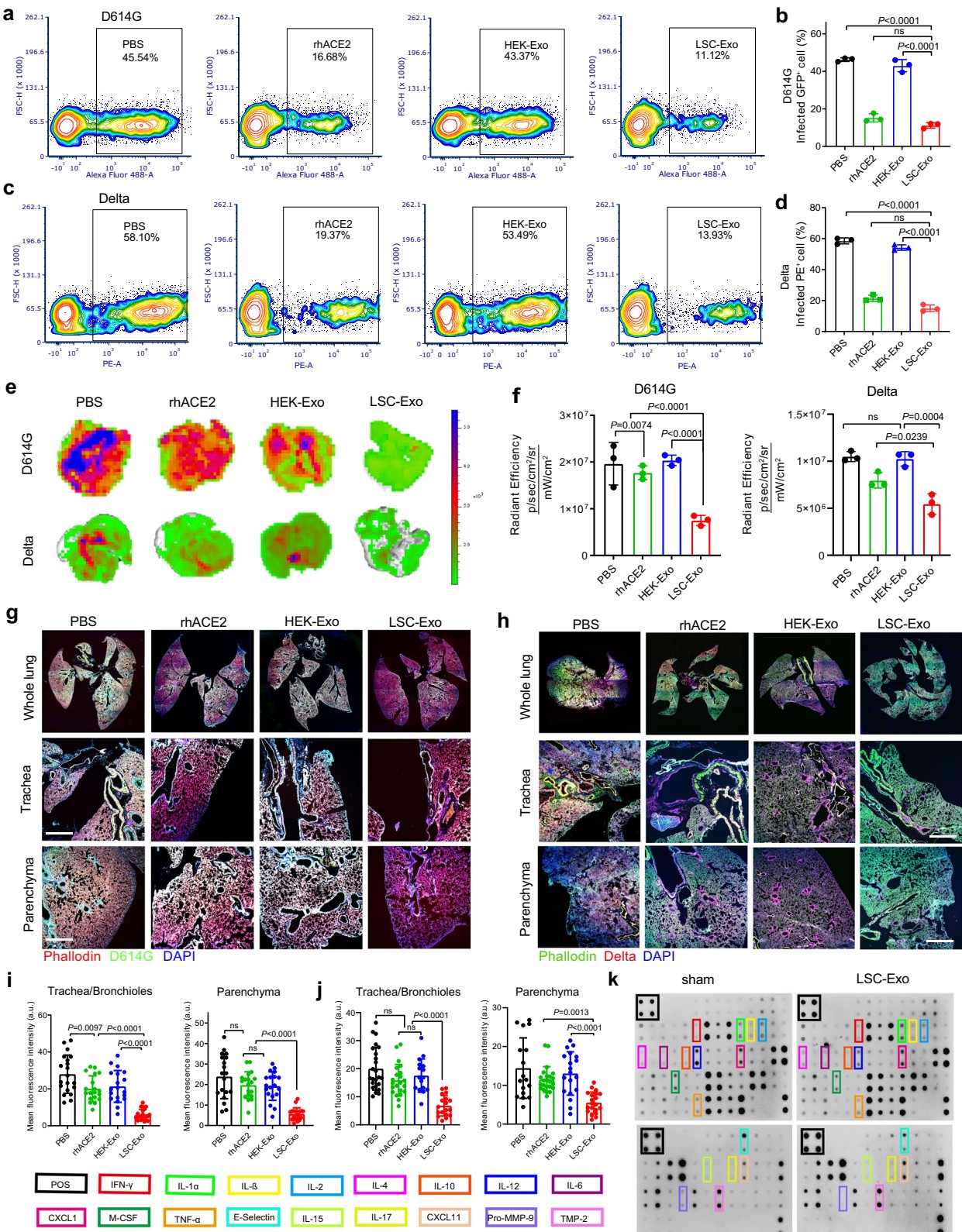

reduce immune activation, maintain intracellular ROS homeostasis, and dampen inflammatory cytokine storm, but also alleviated pulmonary dysfunction in the hamsters by activating the antioxidant defense systems. Despite the emergence of SARS-COV-2 VOC intensively decreased the effectiveness of current vaccines and neutralizing antibodies[54], our datasets demonstrated that LSC-Exo retain potent neutralization activity for all variant pseudoviruses examined,

efficiently intercepting the D614G and B.1.617.2 (Delta) pseudoviruses entry into the lung of mice.

We envision that our ACE2-containing LSC-Exo could serve as a convenient and cost-effective agent to prevent initial infection or further internal dissemination of the virus, reduce viral transmission and alleviate disease onset of COVID-19. While this approach shows promise for clinical translation, several critical issues require careful

**Fig. 6 | LSC-Exo prevents the infection of SARS-CoV-2 D614G and B.1.617.2 (Delta) pseudoviruses. a** Flow cytometry of A549 cells infected with SARS-CoV-2 D614G pseudovirus, which were inhibited by LSC-Exo, HEK-Exo or rhACE2 treatment and the corresponding quantification analysis (**b**). $n = 3$. Gating strategy was shown in Fig. S20b. **c** Flow cytometry of SARS-CoV-2 B.1.617.2 (Delta) pseudovirus-infected A549 cells treated with LSC-Exo, HEK-Exo or rhACE2 and its corresponding quantification analysis (**d**). $n = 3$. Gating strategy was same with Fig. S20b. **e** Ex vivo IVIS imaging of infected lungs from mice with SARS-CoV-2 D614G or B.1.617.2 (Delta) pseudovirus challenge. rhACE2 or HEK-Exo or LSC-Exo was inhaled at 2 h before challenge. **f** Quantitative fluorescence intensity of SARS-CoV-2 D614G or

B.1.617.2 (Delta) pseudoviruses from Fig. 6e. $n = 3$. Confocal images to show the distribution of SARS-CoV-2 D614G pseudovirus (**g**) and B.1.617.2 (Delta) pseudovirus (**h**) in whole lung tissues from the mice with LSC-Exo or HEK-Exo or rhACE2 treatment. Scale bar: 100 μm. Quantitative of pseudoviruses positive signals in both trachea/bronchioles and parenchyma from mice challenged with D614G (**i**) or B.1.617.2 (Delta) (**j**) pseudovirus. $n = 10$ images from 5 hamsters. **k** Cytokine array to determine inflammatory cytokines from mice serum 7 days after LSC-Exo inhalation. Data are mean ± s.d. One-way ANOVA with Bonferroni correction was performed for statistical analysis. Source data are provided as a Source Data file.

consideration. It is essential to establish a harmonized approach to minimize batch-to-batch variation. Implementing rigorous quality control measures at each stage of the manufacturing process is necessary[55]. Maintaining consistency in LSC-Exo production and ensuring homogeneity of their cargo is imperative goals. Moreover, additional steps, such as upscaling conditions, determining the appropriate culture medium for cell growth and expansion, evaluating the need for cell preconditioning, and developing conditioned medium production for LSC-Exo separation, must be addressed[56].

There are several limitations in our current study. Our current study did not assess the protective and therapeutic efficacy of LSC-Exo against authentic H1N1 virus this time. In addition, we only assessed the protective activity of LSC-Exo against SARS-CoV-2 at 2 h post-inhalation and the therapeutic efficacy of LSC-Exo on day 1 after SARS-CoV-2 challenge, longer intervals should be tested in the future. Nonetheless, we have demonstrated that LSC-Exo is effective in vivo against multiple SARS-CoV-2 D614G and B.1.617.2 (Delta) pseudoviruses, indicating that inhalation of LSC-Exo has the potential to protect the public against emerging and more virulent SARS-CoV-2 variants. In summary, LSC-Exo has the potential to be a highly promising daily prophylaxis drug and would simplify the antiviral treatment strategy against diverse SARS-CoV-2 variants, including those that are yet to emerge.

## Methods
### Cell culture
Lung spheroid cells (LSC) were isolated from lung samples of healthy human obtained from the National Disease Research Interchange and passaged every 3–5 days, which has been conducted in our previous studies[32]. After 2–3 passages, LSC were plated on a fibronectin-coated flask and maintained in Iscove's Modified Dulbecco's Media (IMDM) containing 20% fetal bovine serum (FBS). Media changes were performed every other day. LSC were allowed to reach 70–80% confluence before generating serum-free secretome (LSC-Secretome). LSC-Secretomes were collected and filtered with a 0.22 μm filter to remove cellular debris. HEK293T cells (CRL-3216) were purchased from ATCC and cultured in Dulbecco's Modified Eagle Medium (DMEM) with 10% FBS. A549 cells expressing human ACE2 and human TMPRSS2 were purchased from InvivoGen (a549-hace2tpsa) and cultured in DMEM with 10% FBS. All procedures in this study were in accordance with the ethical standards of the institutional research committee and with the guidelines set by the Declaration of Helsinki.

### Exosome isolation and characterization
Exosomes were collected and isolated from LSC-Secretome via the combination of tangential flow filtration (TFF) and ultrafiltration[57–59]. Filtered secretomes were further filtered with 300 kDa, concentrated and washed with Dulbecco's phosphate-buffered saline (DPBS) through a KrosFlo® KR2i TFF system (REPLIGEN, USA). The exosomes were filtered with a 0.22 μm filter to further remove cellular debris. After that, the collected exosomes were pipetted into a 100 kDa Amicon centrifugal filter unit and centrifuged at 4000 g at 4 °C. Once the medium was filtered, the remaining exosomes were collected from the filter and resuspended using DPBS with 25 mM Trehalose for further

analysis. LSC-Exo and HEK-Exo were analyzed by nanoparticle tracking analysis (NTA; NanoSight NS300, Malvern Panalytical, Malvern, UK), western blot, Nanoimager (ONI, San Diego, USA) and Mass spectrometry. To analyze exosomal morphology, LSC-Exo and HEK-Exo were fixed onto copper grids and stained with vanadium negative staining for TEM (JEOL JEM-2000FX, Peabody, MA, USA).

### Western blot
LSC-Exo and HEK-Exo were lysed by RIPA buffer, reduced and denatured using Laemmli sample buffer with β-mercaptoethanol at 98 °C for 10 min. After that, protein samples and molecular ladder were loaded into a 4–20% acrylamide precast Tris-Glycine gel and ran at 100 V until the samples ran out of the wells, followed by a constant voltage of 200 V was performed. Afterwards, transfer to a PVDF membrane was performed at 100 V for one hour. Following three washes, the membrane was blocked with 5% milk in PBS-T for one hour at room temperature (RT) and incubated with ACE2 antibodies (A4612, ABclonal, 1:500) at 4 °C overnight. After another three washes, the PVDF membrane was incubated with HRP-conjugated Goat Anti-Rabbit IgG H&L antibodies (ab6712, Abcam, 1:10000) for 1.5 h at RT. To visualize the blots, PVDF membranes were incubated with ECL western blotting substrate for 1 -3 mins and imaged in a Bio-Rad Imager.

### Flow cytometry of bead-bound exosomes
For staining of ACE2 receptor on exosome surfaces, $5 \times 10^9$ exosomes were suspended in 50 μL of DPBS and incubated with 50 μL of 4 μm aldehyde/sulfate latex beads ($10^6$) for 15 mins at RT and then moved to 4 °C overnight. 100 μL of 200 mM glycine buffer was added to the above solution and incubated for 30 mins to stop the binding of exosomes with beads. After centrifugation and washing, the pellet was blocked with 100 μL of 5% BSA and then stained with ACE2 antibodies (PA5-85139, Invitrogen) for 1 h at RT. After three washes with MACS flow buffer, the bead-bound-exosomes were resuspended with flow buffer with anti-rabbit IgG with Alexa Fluor® 647 (ab150083, Abcam) for 1 h at 4 °C. After that, bead-bound-exosomes were washed three times for subsequent flow cytometry assay, which were conducted with a CytoFLEX flow cytometer (Beckman Coulter, Brea, CA, USA). FCS Express V6 software was used to analyze flow cytometry data.

### Mass spectrometry of LSC-Exo and HEK-Exo
LSC-Exo and HEK-Exo were spiked with 200 fmol of bovine casein per μg of exosome lysate and were then supplemented with SDS to 5%. Samples were then reduced with 10 mM dithiothreitol for 30 min at 80 °C and alkylated with 20 mM iodoacetamide for 45 mins at RT and supplemented with a final concentration of 1.2% phosphoric acid and 328 μL of S-Trap (Protifi) binding buffer (90% methanol, 100 mM triethylammonium bicarbonate (TEAB)). Proteins were trapped on the S-Trap, digested using 20 ng μL$^{-1}$ sequencing grade trypsin (Promega) for 1 h at 47 °C, and eluted using 50 mM TEAB, followed by 0.2% formic acid (FA), and lastly using 50% acetonitrile, 0.2% FA. All samples were then lyophilized to dryness and resuspended in 12 μL 1% trifluoroacetic acid, 2% acetonitrile containing 12.5 fmol μL$^{-1}$ yeast alcohol dehydrogenase.

Mass spectrometry (MS) was performed on 1 μg of each sample, using an MClass UPLC system (Waters Corp) coupled to a Thermo Orbitrap Fusion Lumos high resolution accurate mass tandem mass spectrometer (Thermo) via a nanoelectrospray ionization source. Briefly, the sample was first trapped on a Symmetry C18 20 mm × 180 μm trapping column (5 μL min⁻¹ at 99.9/0.1 v/v water/acetonitrile), after which the analytical separation was performed using a 1.8 μm Acquity HSS T3 C18 75 μm × 250 mm column (Waters) with a 90-min linear gradient of 5–30% acetonitrile with 0.1% FA at a flow rate of 400 nL min⁻¹ with a column temperature of 55 °C. Data collection on the Fusion Lumos mass spectrometer with a FAIMS Pro device was performed for three difference compensation voltages (−40 V, −60 V, −80 V). Within each CV, a data-dependent acquisition (DDA) mode of acquisition with a r = 120,000 (@ m/z 200) full MS scan from m/z 375-1500 with a target AGC value of 4e⁵ ions was performed. MS/MS scans with HCD settings of 30% were acquired in the linear ion trap in "rapid" mode with a target AGC value of 1e⁴ and max fill time of 35 ms. The total cycle time for each CV was 0.66 s, with total cycle times of 2 s between like full MS scans. A 20 s dynamic exclusion was employed to increase depth of coverage.

Raw LC-MS/MS data files were processed in Proteome Discoverer 3.0 (Thermo Scientific) and then submitted to independent Sequest database searches against a *Human* protein database containing both forward (20260 entries) and reverse entries of each protein. Search tolerances were 2 ppm for precursor ions and 0.8 Da for product ions using trypsin specificity with up to two missed cleavages. All searched spectra were imported into Scaffold (v5.3, Proteome Software) and scoring thresholds were set to achieve a peptide false discovery rate of 1% using the PeptideProphet algorithm. Protein groups with at least 2 peptides were accepted. The normalization mode was selected as the total spectrum amount to correct experimental bias. The normalized total spectra counts were used for quantitative analysis.

### Biodistribution of LSC-Exo in mice

All the mouse experiments were approved by the animal ethical committee and experimental procedures were performed in accordance with the guidelines of the Institutional Animal Care and Use Committee (IACUC) at North Carolina State University under protocol # 19-806-B. To track the biodistribution of LSC-Exo in mice after inhalation, red fluorescent protein (RFP, ab268535, Abcam) was loaded into LSC-Exo and commercial liposome particles (300205, Avanti Polar Lipids) via electroporation, yielding RFP-Exo and RFP-Lipo. Briefly, 10 μg of RFP was added to electroporation buffer solution containing 10⁹ LSC-Exo or liposome particles, which were transferred into an ice-cold 0.4 cm Gene Pulser/MicroPulser Electroporation Cuvette. The electroporation cuvette was inserted into the Gene Pulser Xcell™ Total System (Bio-Rad, Hercules, CA, USA) and electroporated under the following conditions: pulse type: square waveforms; voltage: 200 V; pulse length: 10 msec; number of pulses: 5; pulse interval: 1 s. Electroporation buffer and unloaded RFP were removed by ultrafiltration via an Amicon centrifugal filter (Millipore, UFC510096, 100 kDa molecular weight cutoff) followed by three washes with DPBS buffer (10 mM, pH 7.4, 13,000 g) at 4 °C. RFP encapsulation efficiency was calculated by RFP fluorescence at 595 nm with excitation at 547 nm. The RFP encapsulation efficiency of RFP-LSC was calculated to be 20.43%. The corresponding encapsulation efficiency of RFP-Lipo and RFP-HEK were determined to be 24.57% and 19.35%, respectively.

Seven-eight weeks old female CD1 mice (Crl:CD1(ICR)) were purchased from Charles River Laboratory (Wilmington, MA, USA) to perform this biodistribution assay. Mice were housed in pathogen-free facilities at temperatures of 21–24 °C, with 40–60% humidity, under a 12-h light/dark cycle and with unrestricted access to food and water. Mice were nebulized (Pari Trek S Portable 459 Compressor Nebulizer Aerosol System, 047F45-LCS, PARI, Starnberg, Germany) with RFP-LSC in a single dose of 10⁹ particles per mouse or 0.83 × 10⁹ RFP-Lipo per

mouse or 1.05 × 10⁹ RFP-HEK per mouse, where equal RFP amount in RFP-LSC, RFP-HEK and RFP-Lipo were used. Mice were sacrificed at 2 h, 4 h, or 24 h post-nebulization. The collected lungs were imaged by an Xenogen Live Imager (PerkinElmer, Waltham, MA, USA). To study the distribution of RFP-Exo and Lipo-Exo in lung, the collected lung tissues were fixed in 4% PFA and dehydrated with 30% sucrose solution, and then frozen in O.C.T (Tissue-Tek) and cryosectioned (5 μm). Cryosections were permeabilized, blocked with DAKO containing 0.1% saponin for 1 h, and then stained with Phalloidin antibody (Abcam, 1:1000). ProLong Gold Antifade Mountant with 4′,6-diamidino-2-phenylindole (DAPI) were utilized to counterstain nuclei and prevent fluorophore fade. Imaging was performed with the Olympus FLUOVIEW CLSM.

### Biolayer interferometry assay

Biolayer interferometry assays were performed by the ForteBio Octet-RED96 platform. A solution of RBD at a concentration of 25 μg/mL was used to immobilize RBD antigen on amine reactive 2nd generation (AR2G) biosensor tips by immersion in NHS/EDC (300 s @800 g). RBD-bound biosensor tips were washed with PBS-T (pH 7.4) and separately exposed to LSC-Exo (1 mg/mL), HEK-Exo (1 mg/mL), or rhACE2 (50 μg/mL) to measure the baseline (120 s @800 g), association (300 s @800 g), and dissociation (600 s @800 g). Data analysis was performed using the ForteBio Data Analysis software.

### SARS-CoV-2 pseudovirus neutralization assay in vitro

Wild-type SARS-CoV-2 WA1 pseudovirus carrying the GFP reporter (C1110G) was purchased from Montana Molecular. LSC-Exo, HEK-Exo, or rhACE2 at the indicated concentrations were incubated with SARS-CoV-2 WA1 pseudovirus for 30 mins at 37 °C. After incubation, the mixture was added to A549 cells expressing ACE2 and incubated for another 24 h. The GFP signals from infected cells were detected by fluorescence multi-mode microplate (Infinite M Plex, Tecan Inc.). Additionally, the percentage of infected A549 cells was quantified by flow cytometry assay.

### Mouse studies using SARS-CoV-2 pseudovirus

SARS-CoV-2 D614G mutation pseudovirus carrying the GFP reporter (C1120G) was purchased from Montana Molecular. SARS-CoV-2 B.1.617.2 (Delta) pseudovirus was constructed by co-transfecting HEK293T cells with the plasmids of plv-spike-v8 (InvivoGen), pLenti-EF1pluciferase-PGK-RFP-T2A-PURO lentiviral reporter (LR252, ALSTEM), and pspax2 (64586, Addgene) via Lipofectamine 3000 (L3000015, ThermoFisher Scientific). After 48−72 h, B.1.617.2 (Delta) pseudovirus was harvested from the culture medium through centrifugation (1000 g, 10 mins), aliquoted, and stored at −80 °C until used.

Prior to assessing the neutralization ability of LSC-Exo, the CD1 mice were transduced with adenoviral vector expressing hACE2 (Ad5-hACE2, VectorBuilder). After 5 days, LSC-Exo (10¹⁰ per kg of mouse weight), HEK-Exo (10¹⁰ per kg of mouse weight) or rhACE2 (30 μg per kg of mouse weight) were administered via nebulization. After 2 h, each mouse was challenged with SARS-CoV-2 WA1 pseudovirus or D614G pseudovirus or B.1.617.2 (Delta) pseudovirus. Lungs were excised and imaged at 24 h post-challenge with an Xenogen Live Imager and then cryosectioned for evaluating the distribution of SARS-CoV-2 pseudovirus in mouse lung.

### Biodistribution of LSC-Exo in hamster

All hamster studies without authentic SARS-CoV-2 were approved by the animal ethical committee and experimental procedures were performed in accordance with the guidelines of the IACUC at North Carolina State University under protocol # 23-146-01. Twenty male and female Syrian golden hamsters (Envigo), 6−8 weeks old, were purchased from Envigo, and housed in pathogen-free facilities at

temperatures of 21–24 °C, with 40–60% humidity, under a 12-h light/dark cycle and with unrestricted access to food and water. Hamsters were nebulized (Pari Trek S Portable 459 Compressor Nebulizer Aerosol System, 047F45-LCS, PARI, Starnberg, Germany) with RFP-LSC in a single dose of $4 \times 10^9$ particles per hamster. Hamsters were sacrificed at 2 h or 24 h post-nebulization. The collected organs were imaged by an Xenogen Live Imager (PerkinElmer, Waltham, MA, USA). To study the distribution of LSC-Exo, the collected organs were fixed in 4% PFA and dehydrated with 30% sucrose solution, and then frozen in O.C.T (Tissue-Tek) and cryosectioned (5 μm). Cryo-sections were permeabilized, blocked with DAKO containing 0.1% saponin for 1 h. Pro-Long Gold Antifade Mountant with DAPI were utilized to counterstain nuclei and prevent fluorophore fade. Imaging was performed with the Olympus FLUOVIEW CLSM.

## Hamster studies with live SARS-CoV-2

Twenty male and female Syrian golden hamsters (Envigo), 6–8 weeks old, were randomly divided into four groups. All hamsters were housed at Bioqual Inc. Hamsters were administered with PBS or LSC-Exo or HEK-Exo by nebulization ($n = 5$ per group, 3 F/2 M). 2 hours after inhalation, the hamsters were challenged with $1.99 \times 10^4$ 50% tissue culture infective dose ($\text{TCID}_{50}$) of original SARS-CoV-2 WA1 using the intranasal and intratracheal routes (50 μL in each nare). Bronchoalveolar lavage (BAL), oral swabs (OS), and blood were collected at the indicated time. Hamsters were necropsied at day 7 post-challenge. Another hamster group were challenged with $1.99 \times 10^4$ $\text{TCID}_{50}$ of original SARS-CoV-2 WA1 firstly and then three doses of LSC-Exo was inhaled at day 1, 2, and 3 post-challenge. All hamster studies with authentic SARS-CoV-2 were approved by the animal ethical committee of Bioqual Institutional Animal Care and Use Committee (IACUC, 20-091 P) and experimental procedures were performed in compliance with all relevant local, state, and federal regulations.

## SARS-CoV-2 genomic qPCR assay

A QIAsymphony SP (Qiagen, Hilden, Germany) automated sample preparation platform along with a virus/pathogen DSP midi kit and the complex800 protocol were used to extract viral RNA from 800 μL of OS or BAL. A reverse primer specific to the orf1a sequence of SARS-CoV-2 (5′-CGTGCCTACAGTACTCAGAATC-3′) was annealed to the extracted RNA and reverse transcribed into cDNA using SuperScript™ III Reverse Transcriptase along with RNase Out. The resulting cDNA was treated with RNase H and added to a custom 4x TaqMan™ Gene Expression Master Mix containing primers and a fluorescently labeled hydrolysis probe specific for the orf1a sequence of SARS-CoV-2 (forward primer 5′-GTGCTCATGGATGGCTCTATTA-3′, reverse primer 5′-CGTGCCTACAGTACTCAGAATC-3′, probe 5′-/56-FAM/ ACCTACCTT/ZEN/GAAGGTTCTGTTAGAGTG GT/3IABkFQ/-3′). All PCR setup steps were performed using QIAgility instruments (Qiagen). The qPCR was then carried out on a QuantStudio 3 Real-Time PCR System. SARS-CoV-2 genomic (orf1a) RNA copies per reaction were interpolated using quantification cycle data and a serial dilution of a highly characterized custom RNA transcript containing the SARS-CoV-2 orf1a sequence. Mean RNA copies per milliliter were then calculated by applying the assay dilution factor (DF = 11.7). The limit of quantification of this assay is approximately 31 RNA cp/mL ($1.49 \log_{10}$) with 800 μL of sample.

## Histopathology and immunohistochemistry in infected hamsters

Tissues were fixed with 4% PFA for 24 h and transferred to 70% ethanol. The samples were paraffin embedded and the blocks were sectioned at a thickness of 5 μm. Slides were baked for 1 h at 65 °C, deparaffinized in xylene, and rehydrated by a series of graded ethanol to distilled water. Subsequently, the slides were stained with hematoxylin (HSS16, Sigma-Aldrich) and eosin Y (318906, Sigma-Aldrich). Trichrome (HT10516,

Sigma-Aldrich) assay was conducted according to the instructions of the manufacturer. Optical microscopy was performed to analyze these slides. Lung fibrosis was scored using the Ashcroft scale based on H&E staining, which uses a numerical scale from 0 through 8 to grade fibrosis according to previous report[60].

## RNAscope in situ hybridization in hamsters

SARS-CoV-2 anti-sense-specific probe v-nCoV2019-S (ACD Cat. No. 848561) was purchased to target the positive-sense of the Spike sequence, and SARS-CoV-2 v-nCoV2019-S-sense (ACD Cat. No. 845701) was purchased to target the negative-antisense of the Spike sequence. Prior to performing RNAscope assay, slides were first deparaffinized in xylene, rehydrated, and incubated with RNAscope® $H_2O_2$ (ACD Cat. No. 322335) for 10 mins at room temperature, followed by treatment with retrieval in ACD P2 retrieval buffer (ACD Cat. No. 322000) for 15 mins at 98 °C. After that, slides were incubated with protease plus (ACD Cat. No. 322331) for 30 min at 40 °C. Probe hybridization and detection were performed through the RNAscope® 2.5 HD Detection Reagents-RED (ACD Cat. No.322360) according to the instructions of the manufacturer.

## Immunohistochemistry and immunofluorescence staining of hamster lung sections

For SARS-N, MPO, and MX1 IHC staining, dewaxing and rehydration were performed firstly, retrieval was then performed in citrate buffer (AP9003125, Thermo) followed by treatment with 3% $H_2O_2$ in methanol for 10 mins. Slides were permeabilized and blocked with Dako Protein blocking solution (X0909, DAKO) containing 0.1% saponin (47036, Sigma-Aldrich). After that, slides were incubated with primary rabbit anti-SARS-N antibody (Novus, NB100-56576, 1:200), rabbit anti-MPO (Thermo, PA5-16672, 1:200) and anti-MX1 (Millipore Sigma, MABF938, 1:200) for overnight at 4 °C, respectively, followed by goat anti-rabbit HRP secondary antibody (Abcam, ab6721, 1:1000) or goat anti-mouse HRP secondary antibody (Abcam, ab6789, 1:1000) for 1 h at RT, counterstained with hematoxylin and then bluing with 0.25% ammonia water. Quantification of SARS-N, MPO and MX1 positive cell percentage were counted using the National Institutes of Health ImageJ software.

The pretreatments of slides of immunofluorescence assay were the same as for IHC assay, including dewaxing, rehydration, retrieval, and 3% $H_2O_2$ treatment. After that, slides were blocked with 5% BSA for 30 mins followed by 3 rinses with DPBS. Slides were firstly incubated with primary rabbit anti-SARS-N antibody (1:200) overnight at 4 °C, and then incubated with goat anti-rabbit Alexa Fluor®647 (Abcam, ab150080, 1:500), FITC-pan-CK (abcam, ab78478, 1:200) and Alexa Fluor®568-Iba-1 (Abcam, ab221003, 1:200) at RT for 1 h. Finally, slides were mounted with ProLong Gold Antifade Mountant with DAPI and imaged with Olympus FLUOVIEW CLSM.

## RNA-seq assay

The RNA samples from the lung tissues of hamsters were extracted using Trizol reagent (Invitrogen, 15596026) and submitted to LC Sciences. Inc. RNA quantification, purification, and cDNA library preparation and sequencing were performed by LC Sciences. Inc. RNA-seq data were imported and analyzed in R 3.5.2. RNA-seq data generated in this study have been deposited in the NCBI GEO database under accession code GEO: GSE249987.

## Statistical analysis

All quantitative experiments were conducted in triplicate independently. Data were shown as means ± standard deviation. Student's two-tailed, unpaired *t*-test was used to analyze differences between any two groups. Comparisons of more than two groups were determined using one-way ANOVA followed by the post hoc Bonferroni test. Grouped data were determined by two-way ANOVA followed by Tukey post hoc

test for multiple comparisons. $P < 0.05$ was considered statistically significant.

## Data availability

The Proteomics data generated in this study have been deposited in the ProteomoXchange with identifier PXD047542 via PRIDE database. RNA-seq data generated in this study have been deposited in the NCBI GEO database under accession code GEO: GSE249987. The main data supporting the results in this study are available within the paper and its Supplementary Information. Source data was provided as a Source Data file. Source data are provided with this paper.

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

## Acknowledgements

This work was supported by grants from the National Key R&D Program of China (2023YFC2411305 to Z.W.) and the National Natural Science Foundation of China (32371448 to Z.W.).

## Author contributions

Z.W. and K.C. conceived and designed the study. Z.W., K.D.P., S.L. and X.M. performed the in vitro and in vivo experiments. Z.W., S.H., K.D.P. and X.W. interpreted the results. Z.W., X.W. and K.C. wrote the paper. S.H. and J.L. helped with RNA-seq analysis. D.Z. and P.C.D. performed the biodistribution of LSC-Exo in hamster experiments under BSL2. Y.H. helped with proteomics analysis.

## Competing interests

J.L. is an employee of Xsome Biotech Inc. K.C. is a co-founder and equity holder of Xsome. Xsome provided no funding to the study. No other authors declare competing interests.

## Additional information

**Supplementary information** The online version contains Supplementary Material available at https://doi.org/10.1038/s41467-024-45628-x.

