## [Peer Review File · Nature Communications]

Inhalation of ACE2-expressing lung exosomes provides prophylactic protection against SARS-CoV-2Reviewers' Comments:

Reviewer #1:

Remarks to the Author:

In this manuscript, Cheng and co-authors reported a comprehensive study to prevent broad SARS-CoV-2 infections using lung spheroid cell exosomes. The authors developed and synthesized the human lung spheroid cells (LSC)-derived exosomes (LSC-Exo) with high ACE2-expressing, enabling the LSC-Exo bind and neutralize SARS-CoV-2 and protect the host against the SARS-CoV-2 infection. They showed both in vitro and in vivo results that improved neutralizing capacity against pseudoviruses and elucidated the protective mechanisms. Overall, this is an exciting work with important material development and biological findings. Below are minor comments.

1. In Fig.1d, TEM images of exosomes could be improved with higher resolution to visualize particle morphology and size.
2. In Fig. 2, the authors showed the biodistribution of exosomes in the lungs. After nebulization, it would be helpful to assess the biodistribution of exosomes in other organs. What are the concentrations of exosomes and liposomes used in the analysis of in vivo distribution? Additionally, please add a scale bar in Fig. 2d.
3. The authors applied concentration equivalents in Fig. 3d,e to evaluate the effectiveness of neutralization at the cellular level. How many exosomes were employed here?
4. The authors quantified relevant lung tissue markers, such as positive SARS-N cell and lung fibrosis, in Fig. 4i and j. It would be helpful to provide additional details about the procedures and tools used in Methods.
5. Some figures are in low resolution.
6. Please include additional discussion regarding the future clinical translation of this platform.

Reviewer #2:

Remarks to the Author:

The manuscript reports on the use of exosomes derived from human lung spheroid cells to suppress SARS-CoV-2 infection in vitro and in vivo (mouse and hamster model). Exosomes from human LSCs contained more ACE2 than exosomes from HEK cells. DsRed uploaded in LSC exosomes reached the bronchioles more efficiently than dsRed uploaded in liposomes after inhalation of such exosomes or liposomes by mice using a Pari nebulizer. CD1 mice pre-exposed to inhaled LSC exosomes and 2h later infected with GFP expressing Baculovirus pseudotyped with SARS-CoV-2 D614G or luciferase and RFP expressing lentivirus pseudotyped with delta spike showed reduced reporter gene expression compared with mice that had been pretreated with nebulized PBS, recombinant ACE2, and HEK exosomes. Hamsters that were exposed to inhaled LSC exosomes and 2h later challenged with authentic SARS-CoV-2 presented with reduced body weight loss, viral replication and histopathology scores compared with hamsters that had been pre-exposed to PBS. Bulk RNA seq analysis of the lung tissues of the challenged hamsters, revealed a transcriptional pattern for the LSC exosome treated hamster that was closer to that of a sham-treated hamster than for the PBS treated and challenged hamsters.

The authors have a very well established track record in the development of exosome for therapeutic use. Inhalation of ACE2 containing liposomes as a prophylaxis against SARS-CoV-2 infection is novel. Some experiments, however, require additional controls to support the conclusions.

Major remarks:

1. Exosomes were prepared by passing the 0.22 um filtrate of spent cell culture medium over a 100 kDa Amicon filter and detaching "the remaining exosomes" from the filter. The field seems to evolve to the implementation of rigorous separation and quality control methods to isolate extracellular vesicles (see e.g. PMID: 32854228). Mass spectrometry based QC of the prepared exosomes would be recommended.

2. Line 95: ACE2-deficient HEK293T cells: according to figure 1c, f and g, these cells do express ACE2. Please adapt the statement.
3. It is not clear how the mice were exposed to nebulized materials. Was this done with a specially dimensioned snout-fitting nebulizer? It is stated that nebulization was with 10E9 particles per mouse. How was this dose determined?
4. In the comparison of the biodistribution, RFP-loaded HEK cell-derived exosomes should be included as controls. In addition, equal RFP uploading of exosomes and liposomes should be documented.
5. Outbred CD1 mice were used. These mice are not susceptible to SARS-CoV-2 infection because the spike of human SARS-CoV-2 viruses does not recognize mouse ACE2, at least not the D614G or delta spike used here. How do the authors explain the reporter gene expression in control (PBS) treated mice in figure 3 h-k and in figure 6 e-h? In addition, the bar graphs in figure 3h and figure 6e don't appear to correspond with the presented whole lung images in those panels. Please explain.
6. It is difficult to interpret figure 3a. What does the blue wavy line in the ACE2(?) decorated particles represent? Which symbol represents rhACE2? Are there large and small exosomes? Figure 3: the SARS-CoV-2 virion seems to contain a dsRNA genome whereas this is a positive-stranded RNA virus. Figure 3d and e: the X-axis represents concentrations. How was the concentration of the exosome preparations determined?
7. The hamster challenge experiment lacks a control treatment with HEK-derived exosomes.

Other remarks:

1. The publication by Ching et al., (ACE2-containing defensosomes serve as decoys to inhibit SARS-CoV-2 infection, 2022) should be mentioned in the discussion.
2. Line 74: NCT04252167 does not seem to be registered at clinicaltrials.gov.
3. Is the mouse monoclonal antibody that was raised against human MxA cross-reactive with hamster Mx1 (cfr. figure 5 b and c)?
4. Line 281: what is TNF-beta?
5. Figure 1d, legend: Please check the size of the scale bar.
6. Line 394: please replace "mainly" by "only".
7. Line 401: it is not clear that LSC-exo administration alone results in antioxidant activity, cfr. "intrinsic". It appears that transcriptome analysis was performed on SARS-CoV-2 infected hamsters that had been pre-exposed to LSC-exo.
8. Line 454, please specify the SARS-CoV-2 challenge strain that was used for the hamster experiment.

Reviewer #3:

Remarks to the Author:

"Inhalation of ACE2-expressing lung spheroid cell exosomes for prophylaxis of broad SARS-CoV-2 infection", by Dr Chang and colleagues (manuscript number NCOMMS-22-49185)

In this study, the Wang et al investigate the potential of ACE2-expressing human lung spheroid cells (LSC) derived extracellular vesicles to neutralize SARS-CoV-2 and inhibit the infection of multiple SARS-CoV-2 variants. For this, hACE2-expressing LSC-Exo were administered by nebulization. Biodistribution, retention and neutralization efficacy of LSC-Exo was studied in mice, while their prophylactic capacity against SARS-CoV-2 infection was assessed in Syrian hamster. They showed convincingly that ACE2-expressing LSC-Exo, but not ACE2-negative EVs, were able to bind and neutralize SARS-CoV-2 as well as SARS-CoV-2 variants of concern in vitro and in vivo (mice). The prophylactic activity of LSC-Exo against SARS-CoV-2 infection was demonstrated in Syrian hamster and an explanation for the underlying protection mechanisms provided.

The manuscript addresses a timely topic. The presented data are sound and support the conclusions drawn. The authors critically discuss their findings and address the limitations of their study. In summary, this is a very well-conducted scientific study that shows ACE2-expressing extracellular vesicles as an interesting therapeutic approach for the treatment of SARS-CoV-2 infection.

Comments:

1) The biodistribution and neutralization efficacy was assessed in mice but not in Syrian hamsters? Are data available in hamsters as well?

2) Controls in the experiments with Syrian hamster: PBS, instead of ACE2-negative EVs, was used in the control group. Furthermore, a control group with LSC-Exo without subsequent SARS-CoV-2 infection is missing. Other EV cargo (proteins, nucleic acids) may have an effect as well? Human LSC-EVs may affect the immune response/transcriptome in Syrian hamster even in the absence of SARS-CoV-2 infection.

Minor comments:

3) Based on the method of EV isolation, it is unlikely that solely exosomes, which are per definition extracellular vesicles of endosomal origin, were purified. The term small EVs (sEVs) would be more accurate, as recommended by the MISEV guidelines (<https://doi.org/10.1080/20013078.2018.1535750>).

4) Fig. 1d: The TEM images are of poor quality and hardly allow an assessment of the morphology. Additional data on the characteristics and purity of the isolated EVs should be provided according to the MISEV guidelines.

5) Fig. 3d,e: The dose is provided in ng/μL. LSC- and HEK-Exo samples contain other proteins as well and thus are not 'pure' ACE-2 proteins when compared to rhACE2. It is assumed that rhACE2 contains more ACE2 than LSC-Exo. Does the protein concentration correlate with the EV particle concentration? Are LSC-Exo and HEK-Exo comparable in this respect?

6) Fig. 6k: Cytokine arrays from serum of mice treated with rhACE2 and HEK-Exo: HEK-Exo closer to sham or to the LSC-Exo group?

Editorial Note: Figure on page 11 in this Peer Review File has been amended to remove third-party material where no permission to publish could be obtained.

Reviewers' Comments:

Reviewer #1:

Remarks to the Author:

In this manuscript, Cheng and co-authors reported a comprehensive study to prevent broad SARS-CoV-2 infections using lung spheroid cell exosomes. The authors developed and synthesized the human lung spheroid cells (LSC)-derived exosomes (LSC-Exo) with high ACE2-expressing, enabling the LSC-Exo bind and neutralize SARS-CoV-2 and protect the host against the SARS-CoV-2 infection. They showed both in vitro and in vivo results that improved neutralizing capacity against pseudoviruses and elucidated the protective mechanisms. Overall, this is an exciting work with important material development and biological findings. Below are minor comments.

1. In Fig.1d, TEM images of exosomes could be improved with higher resolution to visualize particle morphology and size.
2. In Fig. 2, the authors showed the biodistribution of exosomes in the lungs. After nebulization, it would be helpful to assess the biodistribution of exosomes in other organs. What are the concentrations of exosomes and liposomes used in the analysis of in vivo distribution? Additionally, please add a scale bar in Fig. 2d.
3. The authors applied concentration equivalents in Fig. 3d,e to evaluate the effectiveness of neutralization at the cellular level. How many exosomes were employed here?
4. The authors quantified relevant lung tissue markers, such as positive SARS-N cell and lung fibrosis, in Fig. 4i and j. It would be helpful to provide additional details about the procedures and tools used in Methods.
5. Some figures are in low resolution.
6. Please include additional discussion regarding the future clinical translation of this platform.

Reviewer #2:

Remarks to the Author:

The manuscript reports on the use of exosomes derived from human lung spheroid cells to suppress SARS-CoV-2 infection in vitro and in vivo (mouse and hamster model). Exosomes from human LSCs contained more ACE2 than exosomes from HEK cells. DsRed uploaded in LSC exosomes reached the bronchioles more efficiently than dsRed uploaded in liposomes after inhalation of such exosomes or liposomes by mice using a Pari nebulizer. CD1 mice pre-exposed to inhaled LSC exosomes and 2h later infected with GFP expressing Baculovirus pseudotyped with SARS-CoV-2 D614G or luciferase and RFP expressing lentivirus pseudotyped with delta spike showed reduced reporter gene expression compared with mice that had been pretreated with nebulized PBS, recombinant ACE2, and HEK exosomes. Hamsters that were exposed to inhaled LSC exosomes and 2h later challenged with authentic SARS-CoV-2 presented with reduced body weight loss, viral replication and histopathology scores compared with hamsters that had been pre-exposed to PBS. Bulk RNA seq analysis of the lung tissues of the challenged hamsters, revealed a transcriptional pattern for the LSC exosome treated hamster that was closer to that of a sham-treated hamster than for the PBS treated and challenged hamsters.

The authors have a very well established track record in the development of exosome for therapeutic use. Inhalation of ACE2 containing liposomes as a prophylaxis against SARS-CoV-2 infection is novel. Some experiments, however, require additional controls to support the conclusions.

Major remarks:

1. Exosomes were prepared by passing the 0.22 um filtrate of spent cell culture medium over a 100 kDa Amicon filter and detaching "the remaining exosomes" from the filter. The field seems to evolve to the implementation of rigorous separation and quality control methods to isolate extracellular vesicles (see e.g. PMID: 32854228). Mass spectrometry based QC of the prepared exosomes would be recommended.

2. Line 95: ACE2-deficient HEK293T cells: according to figure 1c, f and g, these cells do express ACE2. Please adapt the statement.
3. It is not clear how the mice were exposed to nebulized materials. Was this done with a specially dimensioned snout-fitting nebulizer? It is stated that nebulization was with 10E9 particles per mouse. How was this dose determined?
4. In the comparison of the biodistribution, RFP-loaded HEK cell-derived exosomes should be included as controls. In addition, equal RFP uploading of exosomes and liposomes should be documented.
5. Outbred CD1 mice were used. These mice are not susceptible to SARS-CoV-2 infection because the spike of human SARS-CoV-2 viruses does not recognize mouse ACE2, at least not the D614G or delta spike used here. How do the authors explain the reporter gene expression in control (PBS) treated mice in figure 3 h-k and in figure 6 e-h? In addition, the bar graphs in figure 3h and figure 6e don't appear to correspond with the presented whole lung images in those panels. Please explain.
6. It is difficult to interpret figure 3a. What does the blue wavy line in the ACE2(?) decorated particles represent? Which symbol represents rhACE2? Are there large and small exosomes? Figure 3: the SARS-CoV-2 virion seems to contain a dsRNA genome whereas this is a positive-stranded RNA virus. Figure 3d and e: the X-axis represents concentrations. How was the concentration of the exosome preparations determined?
7. The hamster challenge experiment lacks a control treatment with HEK-derived exosomes.

Other remarks:

1. The publication by Ching et al., (ACE2-containing defensosomes serve as decoys to inhibit SARS-CoV-2 infection, 2022) should be mentioned in the discussion.
2. Line 74: NCT04252167 does not seem to be registered at clinicaltrials.gov.
3. Is the mouse monoclonal antibody that was raised against human MxA cross-reactive with hamster Mx1 (cfr. figure 5 b and c)?
4. Line 281: what is TNF-beta?
5. Figure 1d, legend: Please check the size of the scale bar.
6. Line 394: please replace "mainly" by "only".
7. Line 401: it is not clear that LSC-exo administration alone results in antioxidant activity, cfr. "intrinsic". It appears that transcriptome analysis was performed on SARS-CoV-2 infected hamsters that had been pre-exposed to LSC-exo.
8. Line 454, please specify the SARS-CoV-2 challenge strain that was used for the hamster experiment.

Reviewer #3:

Remarks to the Author:

"Inhalation of ACE2-expressing lung spheroid cell exosomes for prophylaxis of broad SARS-CoV-2 infection", by Dr Chang and colleagues (manuscript number NCOMMS-22-49185)

In this study, the Wang et al investigate the potential of ACE2-expressing human lung spheroid cells (LSC) derived extracellular vesicles to neutralize SARS-CoV-2 and inhibit the infection of multiple SARS-CoV-2 variants. For this, hACE2-expressing LSC-Exo were administered by nebulization. Biodistribution, retention and neutralization efficacy of LSC-Exo was studied in mice, while their prophylactic capacity against SARS-CoV-2 infection was assessed in Syrian hamster. They showed convincingly that ACE2-expressing LSC-Exo, but not ACE2-negative EVs, were able to bind and neutralize SARS-CoV-2 as well as SARS-CoV-2 variants of concern in vitro and in vivo (mice). The prophylactic activity of LSC-Exo against SARS-CoV-2 infection was demonstrated in Syrian hamster and an explanation for the underlying protection mechanisms provided.

The manuscript addresses a timely topic. The presented data are sound and support the conclusions drawn. The authors critically discuss their findings and address the limitations of their study. In summary, this is a very well-conducted scientific study that shows ACE2-expressing extracellular vesicles as an interesting therapeutic approach for the treatment of SARS-CoV-2 infection.

Comments:

1) The biodistribution and neutralization efficacy was assessed in mice but not in Syrian hamsters? Are data available in hamsters as well?

2) Controls in the experiments with Syrian hamster: PBS, instead of ACE2-negative EVs, was used in the control group. Furthermore, a control group with LSC-Exo without subsequent SARS-CoV-2 infection is missing. Other EV cargo (proteins, nucleic acids) may have an effect as well? Human LSC-EVs may affect the immune response/transcriptome in Syrian hamster even in the absence of SARS-CoV-2 infection.

Minor comments:

3) Based on the method of EV isolation, it is unlikely that solely exosomes, which are per definition extracellular vesicles of endosomal origin, were purified. The term small EVs (sEVs) would be more accurate, as recommended by the MISEV guidelines (<https://doi.org/10.1080/20013078.2018.1535750>).

4) Fig. 1d: The TEM images are of poor quality and hardly allow an assessment of the morphology. Additional data on the characteristics and purity of the isolated EVs should be provided according to the MISEV guidelines.

5) Fig. 3d,e: The dose is provided in ng/μL. LSC- and HEK-Exo samples contain other proteins as well and thus are not 'pure' ACE-2 proteins when compared to rhACE2. It is assumed that rhACE2 contains more ACE2 than LSC-Exo. Does the protein concentration correlate with the EV particle concentration? Are LSC-Exo and HEK-Exo comparable in this respect?

6) Fig. 6k: Cytokine arrays from serum of mice treated with rhACE2 and HEK-Exo: HEK-Exo closer to sham or to the LSC-Exo group?

**Point-by-Point Response**

*Nature Communications manuscript NCOMMS-22-49185*

*Inhalation of ACE2-expressing lung spheroid cell exosomes for prophylaxis of broad SARS-CoV-2*
*infection*

**Reviewer #1**

**In this manuscript, Cheng and co-authors reported a comprehensive study to prevent broad**
**SARS-CoV-2 infections using lung spheroid cell exosomes. The authors developed and**
**synthesized the human lung spheroid cells (LSC)-derived exosomes (LSC-Exo) with high**
**ACE2-expressing, enabling the LSC-Exo bind and neutralize SARS-CoV-2 and protect the host**
**against the SARS-CoV-2 infection. They showed both in vitro and in vivo results that improved**
**neutralizing capacity against pseudoviruses and elucidated the protective mechanisms.**
**Overall, this is an exciting work with important material development and biological findings.**
**Below are minor comments.**

Re: We thank the reviewer for her/his valuable comments that helped us substantially improve our
manuscript. We have addressed each of the comments below and made revisions accordingly.

**1. In Fig.1d, TEM images of exosomes could be improved with higher resolution to visualize**
**particle morphology and size.**

Re: We thank the reviewer for this good suggestion. The high-resolution TEM images of exosomes
were added in the new Fig. 1d.

**Fig. 1d.** TEM images of LSC-Exo and HEK-Exo. Scale bar: 100 μ m.

**2. In Fig. 2, the authors showed the biodistribution of exosomes in the lungs. After**
 **nebulization, it would be helpful to assess the biodistribution of exosomes in other organs.**
 **What are the concentrations of exosomes and liposomes used in the analysis of in vivo**
 **distribution? Additionally, please add a scale bar in Fig. 2d.**

Re: We thank the reviewer for these great suggestions. The biodistribution of exosomes in other
 organs were studied. As shown in the new **Fig. S5**, in addition to lung, we observed that HEK-Exo
 and Liposome signals were starting to be shown in liver, spleen, and kidneys at 4 hours post-
 inhalation (**Fig. S5a, b**). Comparatively, the LSC-Exo signals beginning to appear in major organs at
 24 hours post-inhalation. These results could be attributed to the homologous targeting effects of
 LSC-Exo to lungs (as those exosomes were derived from lung cells). Confocal images of 24 hours
 post-inhalation exhibited that LSC-Exo, HEK-Exo and liposomes were detected in the heart, liver,
 spleen and kidneys (**Fig. S5c**), which might be attributed to the translocation of nanoparticles from
 the pulmonary tree to the bloodstream followed by delivery to other major organs through circulation.

For the concentrations of exosomes and liposomes used in the analysis of in vivo distribution
 depended on the RFP loading within the exosomes and liposomes. In this manuscript, 10^9 LSC-Exo
 40 per mouse was used for the biodistribution analysis in mice. Per to the determination of RFP
 fluorescence at 595 nm, the RFP encapsulation efficiency of RFP-LSC was calculated to be 20.43%.
 The corresponding encapsulation efficiency of RFP-Lipo and RFP-HEK were determined to be 24.57%
 and 19.35%, respectively. As such, 0.83×10^9 RFP-Lipo and 1.05×10^9 RFP-HEK were used for per
 mouse. We have added this information in the revised manuscript. In addition, the scale bar in **Fig.**
 **2d** has been added.

**Fig. S5. Biodistribution of LSC-Exo in mice after inhalation.** (a) Ex vivo imaging of major organs
of mice after RFP-LSC, RFP-HEK or RFP-Lipo inhalation at the indicated time. (b) Quantification of
the integrated density of RFP fluorescence in major organs; $n=3$ per group. (c) Confocal images
showing the biodistribution of RFP-LSC, RFP-HEK or RFP-Lipo in heart, liver, spleen and kidney
tissues and quantitative results from heart, liver, spleen and kidney tissues. Scale bar, 50 μm . Data
are mean \pm s.q. Statistical analysis was performed by one-way ANOVA with Bonferroni correction.

**3. The authors applied concentration equivalents in Fig. 3d,e to evaluate the effectiveness of**
**neutralization at the cellular level. How many exosomes were employed here?**

Re: We thank the reviewer for this question. The neutralization assay was performed at 96-well plates
with 100 μL cell medium, and the highest concentration of used in this assay is 100 $\text{ng}/\mu\text{L}$.
Correspondingly, the numbers of LSC-Exo and HEK-Exo used for the highest concentration were
$3.7\text{e}10^8$ and $4.27\text{e}10^8$, respectively.

**4. The authors quantified relevant lung tissue markers, such as positive SARS-N cell and lung**
**fibrosis, in Fig. 4i and j. It would be helpful to provide additional details about the procedures**
**and tools used in Methods.**

Re: We thank the reviewer for this great suggestion. The procedures and tools for how to quantify the
lung tissue markers were provided in the revised Methods. Here are what we stated:

***Histopathology and immunohistochemistry in infected hamsters***

*Tissues were fixed with 4% PFA for 24 hours and transferred to 70% ethanol. The samples were*
*paraffin embedded and the blocks were sectioned at a thickness of 5 μm . Slides were baked for 1*
*hour at 65 $^{\circ}\text{C}$, deparaffinized in xylene, and rehydrated by a series of graded ethanol to distilled water.*
*Subsequently, the slides were stained with hematoxylin (HSS16, Sigma-Aldrich) and eosin Y (318906,*
*Sigma-Aldrich). Trichrome (HT10516, Sigma-Aldrich) assay was conducted according to the*
*instructions of the manufacturer. Optical microscopy was performed to analyze these slides. Lung*
*fibrosis was scored using the Ashcroft scale based on H&E staining, which uses a numerical scale*
*from 0 through 8 to grade fibrosis according to previous report.⁵²*

***Immunohistochemistry and immunofluorescence staining of hamster lung sections***

*For SARS-N, MPO, and MX1 IHC staining, dewaxing and rehydration were performed firstly, retrieval*
*was then performed in citrate buffer (AP9003125, Thermo) followed by treatment with 3% H_2O_2 in*
*methanol for 10 mins. Slides were permeabilized and blocked with Dako Protein blocking solution*
*(X0909, DAKO) containing 0.1% saponin (47036, Sigma-Aldrich). After that, slides were incubated*
*with primary rabbit anti-SARS-N antibody (Novus, NB100-56576, 1:200), rabbit anti-MPO (Thermo,*
*PA5-16672, 1:200) and anti-MX1 (Millipore Sigma, MABF938, 1:200) for overnight at 4 $^{\circ}\text{C}$,*
*respectively, followed by goat anti-rabbit HRP secondary antibody (Abcam, ab6721, 1:1000) or goat*
*anti-mouse HRP secondary antibody (Abcam, ab6789, 1:1000) for 1 hour at RT, counterstained with*
*hematoxylin and then bluing with 0.25% ammonia water. Quantification of SARS-N, MPO and MX1*
*positive cell percentage were counted using the National Institutes of Health ImageJ software.*

52. Hubner RH, *et al.* Standardized quantification of pulmonary fibrosis in histological samples.
*BioTechniques* **44**, 507-511, 514-507 (2008).

**5. Some figures are in low resolution.**

Re: In the SI, the high-resolution Figures were provided. In the manuscript, we will provide the .ai
format Figures to editor upon publication, ensuring they have high resolution.

**6. Please include additional discussion regarding the future clinical translation of this**
**platform.**

Re: We thank the reviewer for this good suggestion. The additional discussion regarding the future
clinical translation of our platform was provided in the revised manuscript. Here are what we stated:

*We envision that our ACE2-containing LSC-Exo could serve as a convenient and cost-effective agent*
*to prevent initial infection or further internal dissemination of the virus, reduce viral transmission and*
*alleviate disease onset of COVID-19. While this approach shows promise for clinical translation,*
*several critical issues require careful consideration. It is essential to establish a harmonized approach*
*to minimize batch-to-batch variation. Implementing rigorous quality control measures at each stage*
*of the manufacturing process is necessary. Maintaining consistency in LSC-Exo production and*
*ensuring homogeneity of their cargo are imperative goals. Moreover, additional steps, such as*
*upscaling conditions, determining the appropriate culture medium for cell growth and expansion,*
*evaluating the need for cell preconditioning, and developing conditioned medium production for LSC-*
*Exo separation, must be addressed.*

**Reviewer #2**

**The manuscript reports on the use of exosomes derived from human lung spheroid cells to**
**suppress SARS-CoV-2 infection in vitro and in vivo (mouse and hamster model). Exosomes**
**from human LSCs contained more ACE2 than exosomes from HEK cells. DsRed uploaded in**
**LSC exosomes reached the bronchioles more efficiently than dsRed uploaded in liposomes**
**after inhalation of such exosomes or liposomes by mice using a Pari nebulizer. CD1 mice pre-**
**exposed to inhaled LSC exosomes and 2h later infected with GFP expressing Baculovirus**
**pseudotyped with SARS-CoV-2 D614G or luciferase and RFP expressing lentivirus**
**pseudotyped with delta spike showed reduced reporter gene expression compared with mice**
**that had been pretreated with nebulized PBS, recombinant ACE2, and HEK exosomes.**
**Hamsters that were exposed to inhaled LSC exosomes and 2h later challenged with authentic**
**SARS-CoV-2 presented with reduced body weight loss, viral replication and histopathology**
**scores compared with hamsters that had been pre-exposed to PBS. Bulk RNA seq analysis of**
**the lung tissues of the challenged hamsters, revealed a transcriptional pattern for the LSC**
**exosome treated hamster that was closer to that of a sham-treated hamster than for the PBS**
**treated and challenged hamsters.**

**The authors have a very well established track record in the development of exosome for**
**therapeutic use. Inhalation of ACE2 containing liposomes as a prophylaxis against SARS-CoV-**
**2 infection is novel. Some experiments, however, require additional controls to support the**
conclusions.

Re: We thank the reviewer for her/his comments that helped us substantially improve our manuscript.
We have addressed each of the comments below and made revisions accordingly.

**Major remarks:**

**1. Exosomes were prepared by passing the 0.22 um filtrate of spent cell culture medium over**
**a 100 kDa Amicon filter and detaching “the remaining exosomes” from the filter. The field**
**seems to evolve to the implementation of rigorous separation and quality control methods to**
**isolate extracellular vesicles (see e.g. PMID: 32854228). Mass spectrometry based QC of the**
**prepared exosomes would be recommended.**

Re: We thank the reviewer for this great suggestion. According to the separation methods of
extracellular vesicles in PMID: 32854228, the combination of ultrafiltration with TFF methods were
utilized in our manuscript. The detailed experimental process was provided in the revised manuscript.
Here are what we stated:

*Exosomes were collected and isolated from LSC-Secretome via the combination of ultrafiltration with*
*tangential flow filtration (TFF). Filtered secretomes were further filtered with 300 kDa, concentrated*
*and washed with Dulbecco’s phosphate-buffered saline (DPBS) through a KrosFlo® KR2i TFF system*
*(REPLIGEN, USA). The exosomes were filtered with a 0.22 µm filter to further remove cellular debris.*
*After that, the collected exosomes were pipetted into a 100kDa Amicon centrifugal filter unit and*
*centrifuged at 4000g at 4 °C. Once the medium was filtered, the remaining exosomes were collected*
*from the filter and resuspended using DPBS with 25 mM Trehalose for further analysis. LSC-Exo and*
*HEK-Exo were analyzed by nanoparticle tracking analysis (NTA; NanoSight NS300, Malvern*
*Panalytical, Malvern, UK), western blot, Nanoimager (ONI, San Diego, USA) and Mass spectrometry.*
*To analyze exosomal morphology, LSC-Exo and HEK-Exo were fixed onto copper grids and stained*
*with vanadium negative staining for TEM (JEOL JEM-2000FX, Peabody, MA, USA).*

According to the reviewer’s suggestion, we performed mass spectrometry on LSC-Exo and HEK-Exo.
The new data is now available as **Fig. S4**. Venn diagram in **Fig. S4a** revealed that LSC-Exo and HEK-
Exo share 2146 proteins. Further quantitative scatterplots analysis of these shared proteins (**Fig. S4b**)
revealed that the top enriched proteins in LSC-Exo are A2MG, FNC, ALBU etc. We deduced that this
enrichment was due to the use of fibronectin as a coating agent on the cell culture plate for LSC. We
analyzed the exosomal biomarkers and identified the expression of CD9, CD63, CD81, TSG101, Alix,
and VSP36 in both LSC-Exo and HEK-Exo (**Fig. S4c**). GO functional analysis indicated the proteins
of HEK-Exo were mainly involved in RNA and DNA metabolic processes, whereas LSC-Exo were
mainly involved in the extracellular matrix organization, response to growth factor and response to
wounding etc (**Fig. S4d**). KEGG analysis revealed that mRNA surveillance pathway, Ras signaling
pathway and cell cycle etc were enriched in HEK-Exo. In comparison, cytokine-cytokine receptor

interaction, TGF-beta and NOD-like receptor signaling pathway *etc.* were enriched for LSC-Exo (Fig. S4e). These results suggest the successful isolation of LSC-Exo, which meets QC standards and exhibits a higher anti-inflammatory activity compared to HEK-Exo.

Fig. S4. Proteomics analysis of LSC-Exo and HEK-Exo. (a) Venn diagram of proteins identified in LSC-Exo and HEK-Exo. (b) Quantitative scatterplots analysis of shared proteins in LSC-Exo and HEK-Exo. (c) Peptide numbers of specific biomarkers of exosomes. (d) GO function analysis of LSC-Exo and HEK-Exo. (e) KEGG pathway enrichment of LEC-Exo and HEK-Exo.

2. Line 95: ACE2-deficient HEK293T cells: according to figure 1c, f and g, these cells do express ACE2. Please adapt the statement.

Re: We thank the reviewer for this great suggestion. We have revised our statement as below: HEK293T cells with low ACE2 expression.

3. It is not clear how the mice were exposed to nebulized materials. Was this done with a specially dimensioned snout-fitting nebulizer? It is stated that nebulization was with 10E⁹ particles per mouse. How was this dose determined?

Re: We thank the reviewer for these great questions. Pari Trek S Portable 459 Compressor Nebulizer
Aerosol System (047F45-LCS, PARI, Starnberg, Germany) was employed in our manuscript. This
nebulizer is designed for multiple mice instead of being snout-fitted for individual mice, which was
shown in Fig. 1 for reviewer only.

As the reviewer mentioned, the concentration of LSC-Exo nebulized for mice is $10E^9$ particles per
mouse, we dispersed $10E^9$ of LSC-Exo particles into 1 mL PBS buffer and then this nebulizer was
used to deliver LSC-Exo solution in the form of a fine into mouse.

[Redacted]

**Fig. 1 for reviewer only.** The nebulizer box designed for mice.

**4. In the comparison of the biodistribution, RFP-loaded HEK cell-derived exosomes should be**
**included as controls. In addition, equal RFP uploading of exosomes and liposomes should be**
**documented.**

Re: We thank the reviewer for these good suggestions. We added this new control. The RFP-load
HEK cell-derived exosomes (HEK-Exo) control has been prepared and the equal RFP amount used
for each mouse was supplemented in the revised SI. Per to the RFP fluorescence at 595 nm, the RFP
encapsulation efficiency of RFP-LSC was calculated to be 20.43%. The corresponding encapsulation
of RFP in RFP-Lipo and RFP-HEK were determined to be 24.57% and 19.35%, respectively.
Accordingly, 0.83×10^9 RFP-Lipo and 1.05×10^9 RFP-HEK were used for per mouse. The equal RFP
uploading in RFP-LSC, RFP-HEK and RFP-Lipo used was documented in the revised manuscript.

RFP-HEK control has been supplemented in the biodistribution experiments in new **Fig. 2**. We found
that HEK-Exo signals began evident across the whole lungs after 2 hours of inhalation, exhibiting a
distribution pattern similar to that of LSC-Exo at both 2 hours and 4 hours post-inhalation (**Figs, 2b-**
**d**). However, after 24 hours of inhalation, fewer HEK-Exo signals were detected in trachea,

bronchioles and parenchyma of lungs compared to LSC-Exo (Fig. 2e), indicating that LSC-Exo has a prolonged retention time than HEK-Exo.

**New Fig. 2. RFP-loaded LSC-Exo have superior distribution in mouse lung.** (a) Study scheme of
 RFP-Exo's distribution in mice. (b) 2, 4 and 24 hours *ex-vivo* imaging of mouse lungs after RFP-Exo,
 RFP-HEK or RFP-Lipo nebulization. (c) Quantification results of RFP fluorescence intensity in *ex-vivo*
 mouse lungs; *n*=3. (d) Representative immunostaining images for whole lung, trachea, bronchioles,
 and parenchyma. Exosomes (red) or liposomes (red), phalloidin (green) and DAPI (blue). Scale bar,

1000 μm for whole lung, 100 μm for parenchyma. (e) Quantification results of RFP fluorescence density in trachea, bronchioles, and parenchyma; $n=3$ per group. Data are mean \pm s.d. The one-way ANOVA with Bonferroni correction was performed for statistical analysis.

5. Outbred CD1 mice were used. These mice are not susceptible to SARS-CoV-2 infection because the spike of human SARS-CoV-2 viruses does not recognize mouse ACE2, at least not the D614G or delta spike used here. How do the authors explain the reporter gene expression in control (PBS) treated mice in figure 3 h-k and in figure 6 e-h? In addition, the bar graphs in figure 3h and figure 6e don't appear to correspond with the presented whole lung images in those panels. Please explain.

Re: We thank the reviewer for pointing these out. We apologize for not providing enough experimental details. In mice, the CD1 mice were transduced with adenoviral vector expressing hACE2 (Ad5-hACE2, VectorBuilder) firstly. Five days after Ad5-hACE2 transduction, mice were employed to perform the neutralization ability of LSC-Exo against SARS-CoV-2 pseudovirus and variants pseudovirus in mice. We have made it clear in the revised manuscript. As for the second question, we speculate that the mismatch between bar graphs in **Fig. 3h** and **Fig. 6e** with the presented whole lung images, may be attributed to the autofluorescence interference caused by residual blood in the lung tissues following abdominal aortic phlebotomy sacrifice.

6. It is difficult to interpret figure 3a. What does the blue wavy line in the ACE2(?) decorated particles represent? Which symbol represents rhACE2? Are there large and small exosomes? Figure 3: the SARS-CoV-2 virion seems to contain a dsRNA genome whereas this is a positive-stranded RNA virus. Figure 3d and e: the X-axis represents concentrations. How was the concentration of the exosome preparations determined?

Re: In the **Fig. 3a**, the blue wavy line in the LSC-Exo or HEK-Exo represents the enclosed miRNA. The rhACE2 symbol was not shown in **Fig. 3a**. The large exosomes represent those exosomes close to RBD and the small exosomes represent those far from the RBD. In **Fig. 3**, we utilized SARS-CoV-2 pseudovirus to perform cell experiments. This SARS-CoV-2 pseudovirus is a HIV-based lentivirus, which indeed contains a dsRNA genome. In **Fig. 3d, e**, this neutralization assay was performed at 96-well plates with 100 μL cell medium, and the highest concentration of exosomes employed in this assay was 100 $\text{ng}/\mu\text{L}$. In our manuscript, the concentration of 10^{10} exosome is about 270 μg which determined by microBCA protein assay kit, therefore, the 3.7×10^8 exosomes were used for the highest concentration.

7. The hamster challenge experiment lacks a control treatment with HEK-derived exosomes.

Re: We thank the reviewer for this great suggestion. The HEK-derived exosomes (HEK-Exo) control group has been added in the hamster challenge experiment. The results, as depicted in new **Fig. 4** and **Fig. 5**, encompassing weight loss, viral load in both oral swabs and BAL, RNAscope assay, IHC staining of SARS-N, histological analysis and viral load assay of major organs, revealed that HEK-Exo treatment has little effect on decreasing viral load in oral swabs, BAL and major organs of hamsters as well as attenuating severe pneumonia caused by SARS-CoV-2.

**Fig. 4. Protective effect of LSC-Exo against authentic SARS-CoV-2 infection in Syrian hamsters.**
 (a) Time courses of LSC-Exo inhalation, viral challenge, and measurements. (b) Changes in body
 weight of hamsters over 1-week post-challenge. $n=5$. (c) Viral RNA in oral swabs (OS) from hamsters
 treated with LSC-Exo, HEK-Exo or PBS. $n=5$. (d) Viral RNA in bronchoalveolar lavage (BAL) fluid
 from hamsters treated with LSC-Exo, HEK-Exo or PBS at 7 days post-challenge. $n=5$. (e) RNAscope
 images revealing regional distribution and viral RNA levels in hamster lungs. Immunohistochemistry
 analysis of SARS-N protein in lung tissues of hamsters. Scale bar, 50 μm . (f) Quantification analysis
 of positive SARS-N cell percentages in lungs of hamster. $n=15$. (g) H&E images of representative
 lung sections of hamsters. Scale bar, 500 μm . (h) Masson's trichrome staining of lung sections of
 hamsters. Scale bar, 500 μm . (i) Ashcroft scoring analysis of lung fibrosis from challenged hamsters
 that performed blindly. $n=5$. (j) Spider web plot displaying histopathological scoring of lung damage,
 normalized to sham control (green). Viral genomic RNA levels (k) and sgRNA levels (l) in tissues of

hamsters with PBS, HEK-Exo or LSC-Exo treatment. $n=5$. Data are mean \pm s.d. Statistical analysis
 was performed by two-way ANOVA with Tukey's multiple comparisons (b, c, k and l) or two-tailed,
 unpaired Student's t -test (d, f and i).

**New Fig. 5. Protective mechanisms of LSC-Exo against SARS-CoV-2 infection.** (a) SARS-N (red),
 pan-CK (green), Ibc-1 (purple) and DAPI (blue) staining for lung tissues of hamsters. Scale bar, 50

261 μm . (b) Representative MPO and MX1 IHC images from the lung sections of hamsters. Scale bar, 50
262 μm . (c) Quantification analysis of MPO and MX1 positive cells in hamster lungs. $n=10$. Data are mean
\pm s.d. Statistical analysis was measured by two-tailed, unpaired Student's t -test. (d) Principal
component analysis (PCA) comparing the transcriptome of sham hamster and infected hamsters
treated with PBS or LSC-Exo. (e) Sample clustering based on Pearson's correlation of transcriptomes
in lung tissues from sham, PBS and LSC-Exo group. (f) Venn diagram of the gene profiles between
Sham, PBS and LSC-Exo groups. (g) Volcano plots displaying of differential gene expression from
LSC-Exo versus PBS group (red, upregulated genes; blue, downregulated genes). (h) Gene Ontology
(GO) enrichment analysis of downregulated and upregulated genes from comparisons of infected
hamsters treated with LSC-Exo versus PBS. Heatmaps of expression levels of candidate genes in
oxidative phosphorylation (i), cytokine mediated signaling and NK differentiation (j), MAPK pathway
(k) and TGF- β pathway (l) from the LSC-Exo, PBS and sham groups. (m) Functionally grouped
network of enriched ROS-related categories. Each cluster is represented by a different color. (n) Heat
map showing the differential gene expression of LSC-Exo vs PBS vs Sham.

**Other remarks:**

**1. The publication by Ching et al., (ACE2-containing defensosomes serve as decoys to inhibit**
**SARS-CoV-2 infection, 2022) should be mentioned in the discussion.**

Re: We thank the reviewer for this good suggestion. The reference the reviewer mentioned has been
cited and discussed in the section of Discussion. Here are what we stated:

*In contrast, studies have indicated that ACE2-containing defensosomes in bronchoalveolar lavage*
*fluid from critically ill COVID-19 patients was associated with reduced intensive care unit and*
*hospitalization times.⁵⁰*

*50 Ching KL, et al. An ACE2 Triple Decoy that neutralizes SARS-CoV-2 shows enhanced affinity for*
*virus variants. Sci Rep 11, 12740 (2021).*

**2. Line 74: NCT04252167 does not seem to be registered at clinicaltrials.gov.**

Re: We sincerely appreciate the reviewer for pointing this out. We mis-typed the NCT number, the
correct number is NCT04262167, which has been registered at clinicaltrials.gov
(<https://clinicaltrials.gov/ct2/results?cond=&term=NCT04262167&cntry=&state=&city=&dist=>). We
have corrected it in the revised manuscript.

**3. Is the mouse monoclonal antibody that was raised against human MxA cross-reactive with**
**hamster Mx1 (cfr. figure 5 b and c)?**

Re: We thank the reviewer for pointing this out. The monoclonal antibody MxA was utilized based on
previous reports,^{1,2} which exhibited its cross-reactivity with hamster Mx1.

1. Tostanoski LH, et al. Ad26 vaccine protects against SARS-CoV-2 severe clinical disease in
hamsters. Nat Med 26, 1694-1700 (2020).

2. Frere JJ, et al. SARS-CoV-2 infection in hamsters and humans results in lasting and unique
systemic perturbations after recovery. Sci Transl Med 14, eabq3059 (2022).

**4. Line 281: what is TNF-beta?**

Re: We thank the reviewer for pointing this out. TNF-beta should be TGF-beta, we have corrected it
in the revised manuscript.

**5. Figure 1d, legend: Please check the size of the scale bar.**

Re: We thank the reviewer for pointing this out. We have repeated the TEM assay of both LSC-Exo
and HEK-Exo. And we checked the size of the scale bar and make sure it is correct.

**Fig. 1d.** TEM images of LSC-Exo and HEK-Exo. Scale bar: 100 μ m.

**6. Line 394: please replace “mainly” by “only”.**

Re: We thank the reviewer for this good suggestion. We have revised mainly into only in Line 394.

**7. Line 401: it is not clear that LSC-exo administration alone results in antioxidant activity, cfr.**
**“intrinsic”. It appears that transcriptome analysis was performed on SARS-CoV-2 infected**
**hamsters that had been pre-exposed to LSC-exo.**

Re: We thank the reviewer for pointing this out. The reviewer is correct, the transcriptome analysis
was performed on SARS-CoV-2 infected hamsters that had been pre-exposed to LSC-Exo. We have
detected “intrinsic” in the revised manuscript.

**8. Line 454, please specify the SARS-CoV-2 challenge strain that was used for the hamster**
**experiment.**

Re: We thank the reviewer for pointing this out. The original SARS-CoV-2 WA1 strain was used for
the hamster, which has been added in the revised manuscript.

**Reviewer #3**

**In this study, the Wang et al investigate the potential of ACE2-expressing human lung spheroid**
**cells (LSC) derived extracellular vesicles to neutralize SARS-CoV-2 and inhibit the infection of**
**multiple SARS-CoV-2 variants. For this, hACE2-expressing LSC-Exo were administered by**
**nebulization. Biodistribution, retention and neutralization efficacy of LSC-Exo was studied in**
**mice, while their prophylactic capacity against SARS-CoV-2 infection was assessed in Syrian**
**hamster. They showed convincingly that ACE2-expressing LSC-Exo, but not ACE2-negative**
**EVs, were able to bind and neutralize SARS-CoV-2 as well as SARS-CoV-2 variants of concern**
**in vitro and in vivo (mice). The prophylactic activity of LSC-Exo against SARS-CoV-2 infection**
**was demonstrated in Syrian hamster and an explanation for the underlying protection**
**mechanisms provided.**

**The manuscript addresses a timely topic. The presented data are sound and support the**
**conclusions drawn. The authors critically discuss their findings and address the limitations of**
**their study. In summary, this is a very well-conducted scientific study that shows ACE2-**
**expressing extracellular vesicles as an interesting therapeutic approach for the treatment of**
**SARS-CoV-2 infection.**

Re: We thank the reviewer for her/his comments that helped us substantially improve our manuscript.
We have addressed each of the comments below and made revisions accordingly.

**Comments:**

**1. The biodistribution and neutralization efficacy was assessed in mice but not in Syrian**
**hamsters? Are data available in hamsters as well?**

Re: We thank the reviewer for pointing these out. The biodistribution of LSC-Exo in Syrian hamsters
after inhalation were studied. We found that LSC-Exo were predominantly localized in the lungs of
hamsters 2 hours after inhalation (**Fig S7**). In contrast, after 24 hours of inhalation, LSC-Exo exhibited
substantial distribution throughout the major organs of the hamsters. These results were consistent
with the biodistribution results of LSC-Exo in mice.

To evaluate the neutralization capability of LSC-Exo against SARS-CoV-2 in hamsters, we infected
the hamsters by SARS-CoV-2 firstly. After 24 hours, hamsters were inhaled with three doses of LSC-
Exo on days 1, 2, and 3 post-challenge to investigate the potential of LSC-Exo in neutralizing SARS-
CoV-2 WA1 infection (**Fig. S16a**). High levels of SARS-CoV-2 viral load were observed in the OS for
both PBS and LSC-Exo treatment at on 2nd day post-challenge (dpi), whereas a significant decrease
in the viral load in OS was observed in the LSC-Exo group at 4, 7 days dpi compared to PBS group
(**Fig. S16b**). Consistent with OS results, BAL viral load was approximately 4.136 log₁₀ RNA copies
352 per mL in the LSC-Exo group, which was lower than that in the PBS (5.23) group, indicating that LSC-
353 Exo was capable of neutralizing SARS-CoV-2 (**Fig. S16c**). RNAscope analysis demonstrated that
LSC-Exo efficiently repressed the viral replication (**Fig. S16d**). IHC analysis further demonstrated a
pronounced inhibition of SARS-N expression in response to LSC-Exo treatment. (**Fig. S16e**).
Histological analysis revealed that SARS-CoV-2-induced pulmonary hemorrhage and edema, as well

as the significant infiltration of immune cells, were effectively mitigated by LSC-Exo treatment (**Fig. S16f**). Additionally, the levels of viral genomic RNA (**Fig. S16g**) and subgenomic RNA (**Fig. S16h**) in tissues such as the heart, liver, spleen, kidneys, and lymph nodes exhibited a significant reduction in hamsters treated with LSC-Exo. These data suggested that LSC-Exo has the capacity to neutralize SARS-CoV-2 as observed in mice and further block SARS-CoV-2 infection in hamster.

Fig. S7. Biodistribution of LSC-Exo in hamsters after inhalation. (a) Ex vivo imaging of major organs of hamsters 2 hours and 24 hours after RFP-LSC inhalation. (b) Quantification of the integrated density of RFP fluorescence in major organs; $n=3$ per group. (c) Confocal images showing the biodistribution of LSC-Exo in heart, liver, spleen and kidney tissues of hamsters. Scale bar, 50 μm . Data are mean \pm s.d. Statistical analysis was performed by one-way ANOVA with Bonferroni correction.

Fig. S16. Therapeutic efficacy of LSC-Exo against original SARS-CoV-2 WA1 infection in

**hamsters.** (a) Study design of LSC-Exo as a therapeutic agent against SARS-CoV-2 infection. (b)
 Viral RNA in oral swabs from hamsters treated with LSC-Exo or PBS. *n*=5. (c) Viral RNA in BAL fluid
 from hamsters treated with LSC-Exo or PBS at 7 days post-challenge. *n*=5. (d) RNAscope images of
 hamster lungs. Scale bar, 50 μm. (e) Immunohistochemistry analysis and quantification analysis of
 SARS-N protein in lung tissues of hamsters. Scale bar, 50 μm. (f) H&E staining and Masson's
 trichrome images of lung sections of hamsters. Scale bar, 50 μm. Viral genomic RNA levels (g) and
 sgRNA levels (h) in tissues of hamsters with PBS or LSC-Exo treatment. *n*=5. Data are mean ± s.d.

Statistical analysis was performed by two-way ANOVA with Tukey's multiple comparisons (b, g and h) or two-tailed, unpaired Student's *t*-test (c and e).

2. Controls in the experiments with Syrian hamster: PBS, instead of ACE2-negative EVs, was used in the control group. Furthermore, a control group with LSC-Exo without subsequent SARS-CoV-2 infection is missing. Other EV cargo (proteins, nucleic acids) may have an effect as well? Human LSC-EVs may affect the immune response/transcriptome in Syrian hamster even in the absence of SARS-CoV-2 infection.

Re: According to the reviewer's suggestion, the HEK-Exo with low-ACE2 expression as a control group were added in the hamster experiment, as shown in new **Fig. 4**. Based on the results obtained from the weight loss assay, viral load measurements in both oral swabs and BAL, RNAscope assay, IHC staining of SARS-N, histological analysis, and viral load assays conducted on major organs, as presented in the new **Fig. 4**, it is evident that HEK-Exo treatment has minimal impact on reducing viral load in oral swabs, BAL, and major organs of hamsters. And it does not effectively attenuate the severe pneumonia caused by SARS-CoV-2.

Additionally, a control hamster group with LSC-Exo but without SARS-CoV-2 infection was added according to the reviewer's suggestion. The hamsters were inhaled with LSC-Exo and sacrificed 7 days after inhalation. Clinical chemistry and complete blood count (CBC) analyses were performed to assess whether human LSC-Exo could affect the immune response of Syrian hamster when SARS-CoV-2 infection was absent. As depicted in **Fig. S18**, the results indicated there was no significant difference between PBS group and LSC-Exo group in clinical chemistry analysis. Of specific note, our findings revealed that CBC parameters of hamsters inhaled with LSC-Exo remained within normal ranges, but certain CBC parameters, such as white blood cell count, neutrophil and lymphocyte count, were significantly decreased in the LSC-Exo group compared to the PBS group (**Fig. S19**). These results suggest that LSC-Exo possesses anti-inflammatory properties capable of reducing the immune responses in hamsters. Consequently, it is plausible that other exosome cargo (proteins, nucleic acids) may have a little contribution to the protective ability of LSC-Exo against SARS-CoV-2 infection in hamsters by diminishing the inflammatory response caused by SARS-CoV-2.

New Fig. 4. Protective effect of LSC-Exo against authentic SARS-CoV-2 infection in Syrian hamsters.

Changes in body weight of hamsters over 1-week post-challenge. $n=5$. (c) Viral RNA in oral swabs (OS) from hamsters treated with LSC-Exo, HEK-Exo or PBS. $n=5$. (d) Viral RNA in bronchoalveolar lavage (BAL) fluid from hamsters treated with LSC-Exo, HEK-Exo or PBS at 7 days post-challenge. $n=5$. (e) RNAscope images revealing regional distribution and viral RNA levels in hamster lungs. Immunohistochemistry analysis of SARS-N protein in lung tissues of hamsters. Scale bar, 50 μm . (f) Quantification analysis of positive SARS-N cell percentages in lungs of hamster. $n=15$. (g) H&E images of representative lung sections of hamsters. Scale bar, 500 μm . (h) Masson's trichrome staining of lung sections of hamsters. Scale bar, 500 μm . (i) Ashcroft scoring analysis of lung fibrosis from challenged hamsters that performed blindly. $n=5$. (j) Spider web plot displaying histopathological scoring of lung damage, normalized to sham control (green). Viral genomic RNA levels (k) and sgRNA levels (l) in tissues of hamsters with PBS, HEK-Exo or LSC-Exo treatment. $n=5$. Data are mean \pm s.d. Statistical analysis was performed by two-way ANOVA with Tukey's multiple comparisons (b, c, k and l) or two-tailed, unpaired Student's t -test (d, f and i).

**Fig. S18. Clinical chemistry parameters from the peripheral blood of hamsters 7 days after**
 **LSC-Exo inhalation.** Each dot represents data from one animal. Data are mean \pm s.d. $n=3$. Statistical
 analysis the two-tailed, unpaired Student's *t*-test.

**Fig. S19. Complete blood count analysis from the peripheral blood of hamsters 7 days after**
**LSC-Exo inhalation.** Each dot represents data from one animal. Data are mean \pm s.d. $n=3$. Statistical
analysis the two-tailed, unpaired Student's t -test.

**Minor comments:**

**3. Based on the method of EV isolation, it is unlikely that solely exosomes, which are per**
**definition extracellular vesicles of endosomal origin, were purified. The term small EVs (sEVs)**
**would be more accurate, as recommended by the MISEV guidelines**
**(<https://doi.org/10.1080/20013078.2018.1535750>).**

Re: We agreed with the reviewer's opinion that it is unlikely that solely exosomes were purified since
currently available EV purification methods seldom allow for complete separation of exosomes and
ectosomes. Some studies reported that the size of exosomes is about 30-100 nm, however, some
literatures indicated exosomes are small membranous vesicles of 30-150 nm diameter. In our
manuscript, we employed 0.22 μ m filter to remove the cell debris and large extracellular vesicles.
Hence, the reviewer is correct that the small EVs is more accurate than exosomes for our manuscript,
given its physical size characterization.

Recently, multiple studies have been dedicated to discriminating the specific biomarkers between
exosomes and ectosomes. Proteomics analysis demonstrated that VPS24, VPS32 and VPS36 were
exclusively identified in exosomes. In contrast, VSP37D was only detected in ectosomes.¹ We also
performed the proteomics analysis of both LSC-Exo and HEK-Exo by mass spectrometry. We
detected the VPS36 protein, rather than VSP37D protein, indicating that most of extracellular vesicles
we purified are exosomes.

However, this terminology has not yet received official approval from MISEV. Additionally, we
previously published some studies on LSC-derived extracellular vesicles, which were termed as
exosomes (Nat Commun 11(1): 1064, 2020; Nat Biomed Eng 6(7): 791-805, 2022; Matter 5(9): 2960-
2974, 2022; Extracellular Vesicle 1: 100002, 2022). In light of these, we added a statement in the
revised manuscript acknowledging that "small extracellular vesicles (sEVs)" is a more accurate term
than "exosomes." Furthermore, we intend to conduct further investigations to ascertain the presence
of specific biomarkers indicating the endosome-origin of these exosomes. Here are what we stated:

*It is worth noting that the term small extracellular vesicles are considered more accurate than*
*exosomes for characterizing the purified LSC-derived extracellular vesicles according to the MISEV*
*guidelines.*

1. Keerthikumar S, et al. Proteogenomic analysis reveals exosomes are more oncogenic than
ectosomes. *Oncotarget* 6, 15375-15396 (2015).

**4. Fig. 1d: The TEM images are of poor quality and hardly allow an assessment of the**
**morphology. Additional data on the characteristics and purity of the isolated EVs should be**
**provided according to the MISEV guidelines.**

Re: We thank the reviewer for these great suggestions. The high-resolution of TEM images were
provided, as shown in **Fig. 1d**. Both LSC-Exo and HEK-Exo show a cup-shape morphology. It is
recommended by MISEV that at least three positive and one negative protein markers of exosomes
should be conducted for exosome characterization. Three tetraspanins biomarkers were studied by
nanoimager. We found that distinct CD9, CD63 and CD81 biomarkers on single LSC-Exo and HEK-
Exo (**Fig. 1e**). The expression of the cytosolic biomarkers Alix in exosomes have been demonstrated
and acted as a reference protein for studying ACE2 expression on exosomes (**Fig. 1g**). In addition,
the expression of cytosolic marker TSG101 in exosomes compared to negative marker calnexin were
demonstrated (**Fig. S3**).

In addition, we have supplemented all the details regarding our exosome purification methods in the
revised manuscript. Here are what we stated:

**Exosome isolation and characterization**

*Exosomes were collected and isolated from LSC-Secretome via the combination of ultrafiltration with*
*tangential flow filtration (TFF) system. Filtered secretomes were further filtered with 300 kDa,*
*concentrated and washed with Dulbecco's phosphate-buffered saline (DPBS) through a KrosFlo[®]*
*KR2i TFF system (REPLIGEN, USA). The exosomes were filtered with a 0.22 μ m filter to further*
*remove cellular debris. After that, the collected exosomes were pipetted into a 100kDa Amicon*
*centrifugal filter unit and centrifuged at 4000g at 4 °C. Once the medium was filtered, the remaining*
*exosomes were collected from the filter and resuspended using DPBS with 25 mM Trehalose for*
*further analysis. LSC-Exo and HEK-Exo were analyzed by nanoparticle tracking analysis (NTA;*
*NanoSight NS300, Malvern Panalytical, Malvern, UK), western blot and Nanoimager (ONI, San Diego,*
*USA) and Mass spectrometry. To analyze exosomal morphology, LSC-Exo and HEK-Exo were fixed*
*onto copper grids and stained with vanadium negative staining for TEM (JEOL JEM-2000FX, Peabody,*
*MA, USA).*

**Fig. 1d.** TEM images of LSC-Exo and HEK-Exo. Scale bar: 100 μ m.

**Fig. 1e.** 3-color dSTORM image of CD63-Alexa Fluor®-488, CD9-Alexa Fluor®-555, CD81-Alexa
 Fluor®-647 of LSC-Exo (left) or HEK-Exo (right).

**Fig. 1g.** Western blot and quantification analysis of ACE2 levels on LSC-Exo and HEK-Exo. Alix acted
 as the internal control protein.

**Fig. S3.** Western blot of TSG101 and calnexin expression on HEK293T cells, HEK-Exo and LSC-Exo.

**5. Fig. 3d,e:** The dose is provided in ng/μL. LSC- and HEK-Exo samples contain other proteins
 as well and thus are not 'pure' ACE-2 proteins when compared to rhACE2. It is assumed that
 rhACE2 contains more ACE2 than LSC-Exo. Does the protein concentration correlate with the
 EV particle concentration? Are LSC-Exo and HEK-Exo comparable in this respect?

Re: We apologize for this confusion. In the **Figs. 3d,e**, the dose provided in the X-axis referred to the concentration of LSC-Exo and HEK-Exo. And the concentration of rhACE2 used did not shown in this **Figs. 3d,e**. We have added it in the revised manuscript. There are the experiment details:

In our manuscript, the concentration of rhACE2 used is the same with ACE2 amount presented on LSC-Exo. The concentration of 10^{10} LSC-Exo is about 270 μ g, correspondingly, 3.7×10^8 LSC-Exo were used for the highest concentration in **Figs. 3d,e**. ELISA analysis revealed that there are approximately 98 ACE2 receptors per LSC-Exo particle. As a result, 60 ng per mL of rh ACE2 were used for the largest concentration of rhACE2 in the **Figs. 3d,e**. When comparing LSC-Exo and HEK-Exo, we used more HEK-Exo in this assay because 10^{10} HEK-Exo equated to approximately 240 μ g, resulting in the use of 4.27×10^8 HEK-Exo for the highest concentration.

6. Fig. 6k: Cytokine arrays from serum of mice treated with rhACE2 and HEK-Exo: HEK-Exo closer to sham or to the LSC-Exo group?

Re: We thank the reviewer for this question. The cytokine arrays of the mice treated with rhACE2, or HEK-Exo were performed. We found that HEK-Exo group showed a more similarity with LSC-Exo group than sham group, as illustrated in **Fig. S15**.

515

516

517

Fig. S15. Cytokine array to determine inflammatory cytokines from mice serum 7 days after rhACE2 or HEK-Exo or LSC-Exo inhalation.

Reviewers' Comments:

Reviewer #1:

Remarks to the Author:

In this revised manuscript, the authors provided additional description and information to clarify the experiment procedures and data analysis. New experimental data support the experimental design and conclusions.

Reviewer #2:

Remarks to the Author:

The authors have addressed my comments in the revised manuscript and rebuttal very well. I appreciate their efforts, including the additional in vivo experiments that were performed.

I have a few remaining minor remarks:

1. Line 45, "Moreover, mAbs... and line 422, "intrinsic ADE effect of neutralizing antibodies". There is overwhelming evidence that SARS-CoV-2 neutralizing antibodies protect against COVID-19. Please refer to a primary research article to support the statement that SARS-CoV-2 neutralizing mAbs "have a great potential for exacerbation of COVID-19". Alternatively, consider to adapt the statement.
2. Line 471: "H1N1 virus" should likely be replaced by "SARS-CoV-2". If so, please check the sentence because live (better to use "authentic") SARS-CoV-2 challenge experiments were performed in the hamsters.
3. Line 472: ... at 2 hours post inhalation. The protective activity of LSC-exosomes was also tested starting with treatment on day 1 after challenge (Fig. S16). Please check.
4. Line 760: please confirm that the hamster challenge experiments were performed in BSL-2 and not in BSL3.

Reviewer #3:

Remarks to the Author:

The authors have adequately addressed all points raised.

Reviewer #4:

Remarks to the Author:

This manuscript detailing the use of LSC-exosomes as a COVID-19 therapeutic shows a great deal of research effort and supporting evidence. This review is meant to specifically focus on the mass spectrometry data added after the first round of review. Overall, the addition of this data is extremely minimal in context of the paper and adequately addresses the initial reviewer's comments. Specific comments are provided as follows:

- No methods are provided for the mass spectrometry. Though samples were likely sent to a core facility for analysis, the preparation and analysis methods should be included at least in a supplemental methods section.
- Figure S4A: in a Venn diagram the area of the circle should be proportional to the corresponding number
- Figure S4B: I've never heard the term "Quantitative scatterplots analysis". I believe this type of figure is a correlation scatterplot.

Unfortunately, in terms of scientific writing, this manuscript was extremely difficult to read and could greatly benefit from a native English speaker giving it a once-over for grammar and flow. A non-exhaustive list of typos and grammar issues is provided below.

P2L38--- undeniably, typo, are = is, grammar

P2L39--- "ideally providing the prophylaxys"

P2L48--- THE ACE2 hat

P2L51--- inject = injected, grammar

P2L52--- have = has, grammar

P2L73—what is nature mixtures?

L78

110

L118

147

155

175

189

219 - somewhat nonsensical

221 incorrect use of "firstly" which is a very rare term to have a place for proper usage

**Point-by-Point Response**

*Nature Communications manuscript NCOMMS-22-49185A*

*Inhalation of ACE2-expressing lung spheroid cell exosomes provides prophylactic protection against*
*SARS-CoV-2*

**Reviewer #1**

**In this revised manuscript, the authors provided additional description and information to**
**clarify the experiment procedures and data analysis. New experimental data support the**
**experimental design and conclusions.**

Re: We thank the reviewer for her/his good words on our revisions.

**Reviewer #2**

**The authors have addressed my comments in the revised manuscript and rebuttal very well. I**
**appreciate their efforts, including the additional in vivo experiments that were performed. I**
**have a few remaining minor remarks:**

Re: We thank the reviewer for her/his good words on our revisions and thank the comments that
continuously helped us improve our manuscript. We have addressed each of the comments below
and made revisions accordingly.

**Comments:**

**1. Line 45, “Moreover, mAbs... and line 422, “intrinsic ADE effect of neutralizing antibodies”.**
**There is overwhelming evidence that SARS-CoV-2 neutralizing antibodies protect against**
**COVID-19. Please refer to a primary research article to support the statement that SARS-CoV-**
**2 neutralizing mAbs “have a great potential for exacerbation of COVID-19”. Alternatively,**
**consider to adapt the statement.**

Re: We thank the reviewer for this great suggestion. We have adapted the statement in both Line 45
and line 422, Here are what we stated:

*While there is substantial evidence indicating that mAbs did not induce antibody-dependent*
*enhancement (ADE) effect in vivo, recent studies show that several antibodies, such as XG016,*
*XG005, DH1047, DH1041, and MW05, did, indeed, induce ADE, using either pseudoviruses or*
*authentic viruses.*

*Although promising, the rapid degradation of free rhACE2 and the continued emergence of SARS-*
*CoV-2 VOC greatly compromise their therapeutic efficacy.*

**2. Line 471: “H1N1 virus” should likely be replaced by “SARS-CoV-2”. If so, please check the**
**sentence because live (better to use “authentic”) SARS-CoV-2 challenge experiments were**
**performed in the hamsters.**

Re: We are sorry for making this confusion. We want to clarify that we did not study the protective
and therapeutic efficacy of LSC-Exo against authentic H1N1 virus in this manuscript. Hence, H1N1
virus in Line 471 should not be replaced with “SARS-CoV-2”.

**3. Line 472: ... at 2 hours post inhalation. The protective activity of LSC-exosomes was also**
**tested starting with treatment on day 1 after challenge (Fig. S16). Please check.**

Re: We thank the reviewer for pointing this out. We have revised this sentence into “we only assessed
the protective activity of LSC-Exo against SARS-CoV-2 at 2 hours post-inhalation and the therapeutic
efficacy of LSC-Exo on day 1 after SARS-CoV-2 challenge, longer intervals should be tested in the
future.”

**4. Line 760: please confirm that the hamster challenge experiments were performed in BSL-2**
**and not in BSL3.**

Re: We thank the reviewer for this helpful reminder. The hamster experiments with authentic SARS-
CoV-2 challenge were performed in BSL-3, and the biodistribution of LSC-Exo in hamsters were
performed in BSL-2. We have revised this sentence in Line 760 into “D.Z. and P.C.D. performed the
biodistribution of LSC-Exo in hamster experiments under BSL2”.

**Reviewer #3**

**The authors have adequately addressed all points raised.**

Re: We thank the reviewer for her/his comments that helped us substantially improve our manuscript.

**Reviewer #4**

**This manuscript detailing the use of LSC-exosomes as a COVID-19 therapeutic shows a great**
**deal of research effort and supporting evidence. This review is meant to specifically focus on**
**the mass spectrometry data added after the first round of review. Overall, the addition of this**
**data is extremely minimal in context of the paper and adequately addresses the initial**
**reviewer’s comments. Specific comments are provided as follows:**

Re: We thank the reviewer for her/his good words on our mass spectrometry data. We have
addressed each of the comments below and made revisions accordingly.

**Comments:**

**1. No methods are provided for the mass spectrometry. Though samples were likely sent to a**
**core facility for analysis, the preparation and analysis methods should be included at least in**
**a supplemental methods section.**

Re: We thank the reviewer for pointing these out. The preparation and analysis methods of mass
spectrometry were provided in the supplemental methods section. Here are what we stated:

**Mass Spectrometry of LSC-Exo and HEK-Exo**

LSC-Exo and HEK-Exo were spiked with 200 fmol of bovine casein per μg of exosome lysate and
were then supplemented with SDS to 5%. Samples were then reduced with 10 mM dithiothreitol for
30 min at 80 °C and alkylated with 20 mM iodoacetamide for 45 mins at RT and supplemented with a
final concentration of 1.2% phosphoric acid and 328 μL of S-Trap (Protifi) binding buffer (90%
methanol, 100 mM triethylammonium bicarbonate (TEAB)). Proteins were trapped on the S-Trap,
digested using 20 $\text{ng } \mu\text{L}^{-1}$ sequencing grade trypsin (Promega) for 1 h at 47 °C, and eluted using 50
mM TEAB, followed by 0.2% formic acid (FA), and lastly using 50% acetonitrile, 0.2% FA. All samples
were then lyophilized to dryness and resuspended in 12 μL 1% trifluoroacetic acid, 2% acetonitrile
containing 12.5 fmol μL^{-1} yeast alcohol dehydrogenase.

Mass spectrometry (MS) was performed on 1 μg of each sample, using an MClass UPLC system
(Waters Corp) coupled to a Thermo Orbitrap Fusion Lumos high resolution accurate mass tandem
mass spectrometer (Thermo) via a nanoelectrospray ionization source. Briefly, the sample was first
trapped on a Symmetry C18 20 mm \times 180 μm trapping column (5 $\mu\text{L } \text{min}^{-1}$ at 99.9/0.1 v/v
water/acetonitrile), after which the analytical separation was performed using a 1.8 μm Acquity HSS
T3 C18 75 $\mu\text{m} \times$ 250 mm column (Waters) with a 90-min linear gradient of 5–30% acetonitrile with
0.1% FA at a flow rate of 400 $\text{nL } \text{min}^{-1}$ with a column temperature of 55 °C. Data collection on the
Fusion Lumos mass spectrometer with a FAIMS Pro device was performed for three difference
compensation voltages (-40V, -60V, -80V). Within each CV, a data-dependent acquisition (DDA)
mode of acquisition with a $r=120,000$ (@ m/z 200) full MS scan from m/z 375-1500 with a target AGC
value of $4e^5$ ions was performed. MS/MS scans with HCD settings of 30% were acquired in the linear
ion trap in “rapid” mode with a target AGC value of $1e^4$ and max fill time of 35 ms. The total cycle time
for each CV was 0.66 sec, with total cycle times of 2 sec between like full MS scans. A 20 sec dynamic
exclusion was employed to increase depth of coverage.

Raw LC-MS/MS data files were processed in Proteome Discoverer 3.0 (Thermo Scientific) and then
submitted to independent Sequest database searches against a *Human* protein database containing
both forward (20260 entries) and reverse entries of each protein. Search tolerances were 2 ppm for
precursor ions and 0.8 Da for product ions using trypsin specificity with up to two missed cleavages.
All searched spectra were imported into Scaffold (v5.3, Proteome Software) and scoring thresholds
were set to achieve a peptide false discovery rate of 1% using the PeptideProphet algorithm. Protein
groups with at least 2 peptides were accepted. The normalization mode was selected as the total
spectrum amount to correct experimental bias. The normalized total spectra counts were used for
quantitative analysis.

**2. Figure S4A: in a Venn diagram the area of the circle should be proportional to the**
**corresponding number.**

Re: We thank the reviewer for this good suggestion. We have revised Fig. S4A accordingly.

**New Fig. S4a.** Venn diagram of proteins identified in LSC-Exo and HEK-Exo.

**3. Figure S4B: I've never heard the term "Quantitative scatterplots analysis". I believe this type**
**of figure is a correlation scatterplot.**

Re: We thank the reviewer for pointing this out. We have corrected "Quantitative scatterplots analysis"
into the correlation scatterplot in both the Figure S4B legend and Line 112.

**Unfortunately, in terms of scientific writing, this manuscript was extremely difficult to read and**
**could greatly benefit from a native English speaker giving it a once-over for grammar and flow.**
**A non-exhaustive list of typos and grammar issues is provided below.**

Re: We thank the reviewer for this good suggestion. Our manuscript has been revised by a native
English speaker. Those typos and grammar issues the reviewer mentioned have been revised.

**P2L38--- undeniably, typo, are = is, grammar**

Re: We have corrected "are" into "is".

**P2L39--- "ideally providing the prophylaxys"**

Re: We have corrected into "ideally providing prophylaxis at the virus entry portal".

**P2L48--- THE ACE2 hat**

Re: We have revised "ACE2 hat" into "ACE2 receptor".

**P2L51--- inject = injected, grammar**

Re: We have corrected "inject" into "injected".

**P2L52--- have = has, grammar**

Re: We have corrected “have” into “has”.

**P2L73—what is nature mixtures?**

Re: Lung spheroid cells were established from adult lung tissues that were plated on fibronectin-
coated petri dishes to outgrow. The outgrowth cells were collected and plated into low attachment
flasks for the formation of lung spheroids. The collected Lung spheroids were replated onto
fibronectin-coated flasks to dissociate into LSC. We established that LSC contain a heterogeneous
population of cells expressing lung epithelial (Epcam, AQP5 and ProSPC) and mesenchymal (CD90
and CD105) markers. Therefore, the natural mixtures means both lung progenitors and supporting
stromal cells.

**L78**

Re: We have checked the Line 78 and fixed the grammar issue. We replaced “basically starts” with
“typically begins”.

**110**

Re: We have replaced “quantitative” into “correlation”.

**L118**

Re: We have checked the Line 118 and replaced “of” with “from” and “assay” with “assays”.

**147**

Re: We have checked the Line 147 and replaced “across” with “throughout”.

**155**

Re: We have checked the Line 155 and replaced “were starting to be shown” with “began to appear”.

**175**

Re: We have checked the Line 175 and replaced “the specific binding of RBD with rhACE2 was
inhibited by LSC-Exo” with “LSC-Exo inhibited the specific binding of RBD with rhACE2”.

**189**

Re: We have checked the Line 189 and replaced “the mice were nebulized” with “mice were
nebulized”.

**219 – somewhat nonsensical**

Re: We thank the reviewer for pointing this out. We have revised the sentence in Line 219 into Syrian
golden hamsters, exhibiting diverse pathologies characteristic of SARS-CoV-2 infection,³⁴ were
utilized to evaluate the prophylactic and therapeutic capacity of LSC-Exo against SARS-CoV-2
infection.

**221 incorrect use of “firstly” which is a very rare term to have a place for proper usage**

Re: We thank the reviewer for pointing this out. We have deleted “firstly” in Line 221.